# Towards Fast LLM Fine-tuning through Zeroth-Order Optimization with Projected Gradient-Aligned Perturbations

## Abstract

Fine-tuning large language models (LLMs) using zeroth-order (ZO) optimization has emerged as a promising alternative to traditional gradient-based methods due to its reduced memory footprint requirement. However, existing ZO methods suffer from high variance in gradient estimation, leading to slow convergence and suboptimal performance on large-scale models. In this work, we propose P-GAP, a fast LLM fine-tuning approach through zeroth-order optimization with Projected Gradient-Aligned Perturbations. Specifically, we first estimate a low-dimensional gradient space and then align perturbations in projected gradients' direction within the space. This approach enables reduced the number of perturbed parameters and decreased variance, therefore accelerated convergence for LLM fine-tuning. Experiments on LLMs show that P-GAP consistently surpasses the baselines, achieving up to 6% increase in accuracy on classification tasks and up to 12% higher accuracy on generation tasks, with up to about 81% less training iterations and 70% less GPU hours. These results demonstrate that P-GAP enables fast, scalable, and resource-efficient ZO LLM fine-tuning.

## 1 Introduction

Fine-tuning (FT) large language models (LLMs) (Hu et al., 2021; Dettmers et al., 2023; Gu et al., 2021) for specific tasks or datasets has become a common practice in modern machine learning. However, as model size and complexity scale, fine-tuning incurs substantial memory overhead, which severely limits its scalability and makes it inaccessible to users with constrained computational resources (Tan et al., 2025b; Zhao et al., 2024b). To alleviate this issue, parameter-efficient fine-tuning (PEFT) methods have been proposed (Li & Liang, 2021; Dettmers et al., 2023; Zhao et al., 2024a), which update only a small subset of parameters while freezing the majority of the model. These approaches drastically reduce GPU memory footprint and storage cost while achieving performance comparable to full FT. However, despite their efficiency, PEFT methods still require computing and storing full gradients and intermediate activations during backpropagation, which introduces significant memory overhead (Malladi et al., 2023; Liu et al., 2024b).

To address the challenge, zeroth-order (ZO) optimization has emerged as a promising solution (Zhang et al., 2024b; Malladi et al., 2023), which estimates gradients using only forward passes. By leveraging randomized perturbations to approximate gradient directions, ZO completely removes the need to store large gradient tensors and intermediate activations, which substantially reduces memory usage. This advantage makes ZO especially appealing for extremely large models where backward passes dominate GPU memory consumption. When combined with parameter-efficient strategies, ZO-based fine-tuning offers a scalable and resource-friendly framework for adapting high-capacity models under tight memory constraints while maintaining competitive performance (Tan et al., 2025b). Despite the advantages of zeroth-order optimization in reducing memory overhead, these benefits often come at the expense of longer computational time (e.g., GPU hours) and decreased accuracy compared to first-order approaches (Li et al., 2024; Gautam et al., 2024).

Existing works show that variance in the zeroth-order gradient estimation, attributing to the random perturbations, can be a factor for the longer computational time Chen et al. (2024); Park et al. (2025). The larger variance in the estimation of the ZO gradient can lead to suboptimal accuracy and slower

convergence rates compared to first-order methods, making ZO-based fine-tuning less stable and resource-intensive (Kornilov et al., 2023; Zhang et al., 2024b; Lobanov & Gasnikov, 2023). Existing works in LLM fine-tuning such as (Ohta et al., 2020) and (Malladi et al., 2023) aim to reduce the variance via increasing the number of perturbations, which will lead to prolong training time.

Inspired by (Ma & Huang, 2025; Kozak et al., 2023) which find anisotropic perturbations (i.e., the magnitude of perturbations is larger along certain directions and is smaller along others, rather than being uniform in all directions) can potentially help relieve the variance issue in ZO optimization theoretically, we raise the following question:

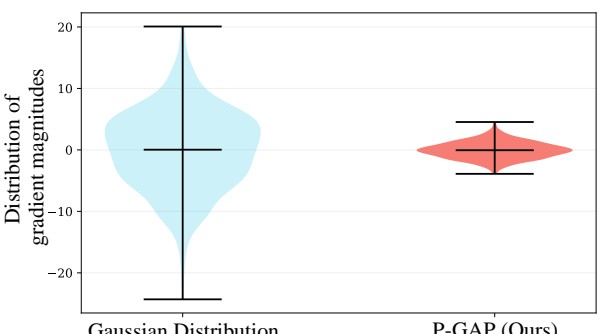

Figure 1: Estimation of directional derivative magnitudes on the $V$ matrix from the first Transformer layer of the OPT-2.7B model, using perturbations sampled from a standard Gaussian distribution and from P-GAP

**Q1:** *For LLM finetuning on larger-scale models, can we find the proper perturbation directions, thereby reducing the variance of ZO gradient estimation and finally accelerating convergence with negligible accuracy loss?*

Inspired by (Wang et al., 2018; Zhang et al., 2024a; Yue et al., 2023), which identify that perturbing the full parameter space can further amplify the variance in gradient estimation, as the variance scales proportionally with the parameter dimension $d$, we naturally pose a research question:

**Q2:** *Can we further reduce variance in gradient estimation by decreasing the parameter space that require perturbation-based gradient estimation?*

To answer the two questions, we propose a fast LLM finetuing approach through zeroth-order optimization with Projected Gradient-Aligned Perturbations (P-GAP), which reduces the variance in gradient estimation of ZO updates via low-dimensional perturbations that are aligned with the gradient direction in the subspace of gradient, thereby achieving faster convergence. Figure 1 shows the magnitude of gradient estimation on the attentions matrix $V$ in the first Transformer layer of the OPT-2.7B model, based on perturbations sampled from a standard Gaussian distribution and from P-GAP. It can be observed that the value of estimated gradients are more stable and less dispersed for P-GAP, which indicates a smaller variance. Our contributions can be summarized as follows:

- We propose a novel ZO-based LLM fine-tuning framework, P-GAP, which estimates a low-dimensional gradient space and aligns perturbations in projected gradients' direction within the space. This design can not only allow the perturbation aligned in the most informative direction but also effectively reduce the dimensionality of gradient estimation, therefore reducing variance and accelerate convergence.

- We provide theoretical analysis on that the variance of ZO gradient estimation linearly increases with the dimension of weight matrix which need perturbations for gradient estimation in LLMs, and further show that P-GAP can reduce the variance with the proposed low-dimensional gradient space estimation. Moreover, we provide the convergence analysis of P-GAP.

- We conduct extensive experiments on both encoder-only models (e.g., RoBERTa-large) and decoder-based LLMs (e.g., OPT-2.7B/6.7B and LLaMA-3-3B/8B). Results show that P-GAP achieves up to 6% accuracy gains over the baselines, while achieving $5.2\times$ speedup in training and more than 61 minutes less wall-clock time.

## 2 PRELIMINARIES

**Notations.** In this paper, all of the non-bold letters (including Latin letters and Greek letters) indicate the scalar such as $\delta$ and $K$. All of the lower-case letters which is bold indicate a column vector such as $\boldsymbol{u}$ and all of the upper-case bold letters such as $\boldsymbol{V}$ indicate a matrix. A $d$-dimensional multivariate Gaussian distribution is denoted by $\mathcal{N}(\boldsymbol{\mu}, \boldsymbol{\Sigma})$, where $\boldsymbol{\mu} \in \mathbb{R}^d$ is the mean vector and $\boldsymbol{\Sigma} \in \mathbb{R}^{d \times d}$ is

the covariance matrix. We use $\mathbb{E}[\cdot]$ to represent the expected value of a variable and use $\text{Var}[\cdot]$ to represent the variance of a variable. $\text{vec}(\boldsymbol{W})$ indicates that we flatten the matrix $\boldsymbol{W}$ by stacking its columns vertically to change it into a column vector. $\|\boldsymbol{x}\|_p = (\sum_{i=1}^n x_i^p)^{\frac{1}{p}}$ indicates the $\ell_p$-norm of a vector $\boldsymbol{x}$ and we use $\|\boldsymbol{x}\|$ to denote the $\ell_2$-norm of a vector $\boldsymbol{x}$. $\|\boldsymbol{U}\|_F = \sqrt{\langle \boldsymbol{U}, \boldsymbol{U} \rangle}$ denotes the Frobenius norm of a matrix $\boldsymbol{U}$ and we will call it F-norm in the paper for simplicity. $\mathcal{C}_L^{s,p}(\mathcal{S})$ denotes the collection of functions defined on the set $S$ that are $s$-times continuously differentiable, and whose $p$-th order derivatives are $L$-Lipschitz continuous. $\widehat{\nabla}$ indicates the estimation of gradient and $\nabla$ indicates the true gradient. $I$ indicates the identity matrix or vector.

**Zeroth-order Optimization for LLMs.** Consider a large language model with parameters $\boldsymbol{\theta} \in \mathbb{R}^d$ and loss function $\mathcal{L}$. At iteration step $t$, zeroth-order optimization estimates the gradient on a mini-batch datasets $\mathcal{B}_t$ by perturbing $\boldsymbol{\theta}_t$ along random directions. Specifically, if we choose to use Gaussian distribution as perturbations, then we can get $\boldsymbol{u} \sim \mathcal{N}(0, I_d)$ and $\mathcal{N}(0, I_d)$ is the standard Gaussian distribution. Given a perturbation scale $\epsilon > 0$, the two-point gradient estimator is

$$\widehat{\nabla}\mathcal{L}(\boldsymbol{\theta}_t; \mathcal{B}_t) = \frac{\mathcal{L}(\boldsymbol{\theta}_t + \epsilon\boldsymbol{u}; \mathcal{B}_t) - \mathcal{L}(\boldsymbol{\theta}_t - \epsilon\boldsymbol{u}; \mathcal{B}_t)}{2\epsilon} \boldsymbol{u} \tag{1}$$

where $\widehat{\nabla}$ in Equation 1 indicates the estimated gradients. To reduce estimator variance, one may average over $n$ independent perturbations $\{\boldsymbol{u}_i\}_{i=1}^n$:

$$\widehat{\nabla}\mathcal{L}(\boldsymbol{\theta}_t; \mathcal{B}_t) = \frac{1}{n} \sum_{i=1}^n \left[ \frac{\mathcal{L}(\boldsymbol{\theta}_t + \epsilon\boldsymbol{u}_i; \mathcal{B}_t) - \mathcal{L}(\boldsymbol{\theta}_t - \epsilon\boldsymbol{u}_i; \mathcal{B}_t)}{2\epsilon} \boldsymbol{u}_i \right] \tag{2}$$

Finally, given the learning rate $\eta$ and estimated gradients in Equation 2, the parameter update follows the standard SGD form:

$$\boldsymbol{\theta}_{t+1} = \boldsymbol{\theta}_t - \eta \widehat{\nabla}\mathcal{L}(\boldsymbol{\theta}_t; \mathcal{B}_t). \tag{3}$$

## 3 METHODOLOGY

In this section, we first clarify the remaining problems in existing zeroth-order optimization frameworks and put up the motivation for our proposed method. Then, we will elaborate on our proposed **P-GAP**, which performs ZO updates with low-dimensional perturbations that are aligned with the gradient direction in the subspace of the gradient for variance reduction. Intuitively, our pipeline begins by obtaining an approximate gradient matrix, which can be expressed as the product of low-rank frame matrices and a coefficient matrix. Within this lower-dimensional space spanned by the frame matrices, Gaussian perturbations may be selected arbitrarily without restriction; however, we hope that they are constrained to be aligned with the directions defined by the gradient's coefficient matrix (i.e. the hyperplane defined by the low-dimension gradient's coefficient matrix). After correction, the perturbation itself can also be represented as a corrected coefficient matrix, which, when multiplied with the frame matrices, yields the final perturbation in the original high-dimensional parameter space. In other words, we allow perturbations to be chosen freely within the linear subspace spanned by low-rank frame matrice, but enforce that they remain parallel to the hyperplane determined by the gradient's coefficient matrix.

### 3.1 PROJECTED GRADIENT-ALIGNED PERTURBATION

Inspired by (Ma & Huang, 2025), we adopt the idea of projecting the sampled random perturbations onto the gradient direction. However, since the original method was designed for the vector dimension, that is, if we generate a random initial perturbation $\boldsymbol{z} \sim \mathcal{N}(0, I_d)$, we hope that the perturbation could satisfy the condition that:

$$(\nabla\mathcal{L}^T \boldsymbol{z})^2 = \delta\|\nabla\mathcal{L}\|^2 \tag{4}$$

which can be simplified to:

$$\langle \nabla\mathcal{L}, \boldsymbol{z} \rangle = \xi \cdot \sqrt{\delta}\|\nabla\mathcal{L}\| \tag{5}$$

where $\xi$ is a constant that is randomly selected from the set $\{-1, 1\}$. And $\langle \cdot, \cdot \rangle$ indicates the inner product of two vectors. However, directly generating the perturbation vector corresponding to Equation 4 and 5 is difficult since it requires sampling from a constrained space rather than the free full parameter space. Since Equation 5 corresponds to a hyperplane in the vector space, we can

randomly sample an initial perturbed vector $\boldsymbol{v}_{init}$ which can be decomposed into two components: one parallel with the hyperplane and the other orthogonal to it. We can denote them as $\boldsymbol{v}_{init\parallel}$ and $\boldsymbol{v}_{init\perp}$, respectively. Then, we only need to retain the parallel component $\boldsymbol{v} = \boldsymbol{v}_{init\parallel}$, which satisfies the requirement of Equation 5. According to (Ma & Huang, 2025), we can calculate the parallel component $\boldsymbol{v}$ of the initial perturbation $\boldsymbol{v}_{init}$ as follows:

$$\boldsymbol{v} = \boldsymbol{v}_{init} - \frac{\nabla\mathcal{L}^T \boldsymbol{v}_{init} - \xi\sqrt{\delta}\,\|\nabla\mathcal{L}\|}{\|\nabla\mathcal{L}\|^2}\,\nabla\mathcal{L} \tag{6}$$

In Equation 6 the aligned perturbation $\boldsymbol{v}$ is not only consistent with the gradient direction and but also satisfies the Gaussian distribution condition, satisfying the following requirements for the chosen perturbations to reduce the variance of ZO gradient estimation (Ma & Huang, 2025; Liu et al., 2020; Gao & Sener, 2022):

- **(a) Constant Magnitude**: The magnitude ($\ell_2$ norm) of the perturbation vector $\boldsymbol{v}$ is a fixed constant, i.e., $\|\boldsymbol{v}\|^2 = d\delta$ ($\delta$ is a random constant). Many traditional methods fall into this category, such as Gaussian distribution, Rademacher distribution and uniform distribution.
- **(b) Directional Alignment**: The square of the inner product between the perturbation vector $\boldsymbol{v}$ and the true gradient $\nabla\mathcal{L}$ is a fixed value, i.e., $(\nabla\mathcal{L}^T \boldsymbol{v})^2 = \delta\|\nabla\mathcal{L}\|^2$. This condition implies that the perturbation direction should be 'aligned' with the gradient direction.

We now extend this theory to the case of high-dimensional matrices. The vector norm on the right-hand side of Equation 5 can be naturally generalized to the matrix norm. In this paper, we adopt the Frobenius norm for matrices, i.e. $\|\boldsymbol{A}\|_F = \sqrt{\sum_{i,j} a_{ij}^2}$, where $a_{ij}$ is the number in the $i$-th row and $j$-th column of the matrix $\boldsymbol{A}$. We can replace the vector inner product with the Frobenius inner product for matrices without loss of generality. For two matrices $\boldsymbol{A}, \boldsymbol{B} \in \mathbb{R}^{m \times n}$, we define

$$\langle \boldsymbol{A}, \boldsymbol{B} \rangle_F = \mathrm{Tr}(\boldsymbol{A}^\top \boldsymbol{B}) \tag{7}$$

where $\mathrm{Tr}(\cdot)$ in Equation 7 means the trace of a matrix and $b_{ij}$ is the number in the $i$-th row and $j$-th column of the matrix $\boldsymbol{B}$. Therefore, the vector hyperplane in Equation 5 can be extended to a tensor hyperplane:

$$\langle \nabla\mathcal{L}, \boldsymbol{Z} \rangle_F = \xi \cdot \sqrt{\delta}\,\|\nabla\mathcal{L}\|_F \tag{8}$$

where $\boldsymbol{Z}$ is a random perturbation satisfying Gaussian distribution.

Similarly, if we randomly generate an initial perturbation matrix $\boldsymbol{C}_{init} \sim \mathcal{N}(0, I_{m \times n})$ and $\boldsymbol{C}_{init}$ have equivalent dimension with gradient matrix $\nabla\mathcal{L} \in \mathbb{R}^{m \times n}$, then the sampled initial perturbation can also be decomposed into a parallel component ($\boldsymbol{C}_{init\parallel}$) and a vertical component ($\boldsymbol{C}_{init\perp}$). Deriving from Equation 6, the parallel component $\boldsymbol{C} = \boldsymbol{C}_{init\parallel}$ of $\boldsymbol{C}_{init}$ can be formulated as:

$$\boldsymbol{C} = \boldsymbol{C}_{init} - \frac{\langle \nabla\mathcal{L}, \boldsymbol{C}_{init} \rangle_F - \xi\sqrt{\delta}\,\|\nabla\mathcal{L}\|_F}{\|\nabla\mathcal{L}\|_F^2}\,\nabla\mathcal{L} \tag{9}$$

We only need to retain the parallel component $\boldsymbol{C}$ of the hyperplane in Equation 8, i.e., the one aligned with the gradient direction and the subsequent ZO perturbation update is then performed using only the parallel component.

## 3.2 Low-dimensional Gradient Space Design

**Motivation.** If we directly apply Equation 9 for perturbation alignment, there are two issues: First, for large language models such as OPT-6.7B, the Transformer layer matrices are very large (e.g., 4096×4096), which leads to high computational cost. Second, Equation 9 still performs perturbation alignment in the full parameter space. However, as we have shown in the Appendix B.1, the larger the dimensionality of the perturbations, the higher the variance of the ZO gradient estimation. This motivates us to explore whether it is possible to restrict the perturbations to a low-dimensional space and perform the perturbation alignment with gradient direction within this low-dimensional space.

Suppose the gradient matrix is denoted as $\boldsymbol{S} = \nabla\mathcal{L} \in \mathbb{R}^{m \times n}$, it can be decomposed in the format of the product of an orthogonal basis matrix and a coefficient matrix, using techniques such as

singular value decomposition (SVD) or QR decomposition. In this work, we adopt SVD for low-rank decomposition, then we have:

$$\boldsymbol{S} \simeq \boldsymbol{U}_r \boldsymbol{S}_r \boldsymbol{V}_r^T \tag{10}$$

where $\boldsymbol{U}_r \in \mathbb{R}^{m \times r}, \boldsymbol{S}_r \in \mathbb{R}^{r \times r}, \boldsymbol{V}_r \in \mathbb{R}^{n \times r}$, $r \ll m$ and $r \ll n$. Evidently, $\boldsymbol{U}_r$ and $\boldsymbol{V}_r$ can be regarded as a pair of frames, i.e., two orthogonal bases. And $\boldsymbol{S}_r$ serves as the set of scaling factors associated with the bases, which indicates the importance of the direction of each singular vector. Hence, a natural choice is to preserve the leading $r$ directions, which captures the most significant components. Then, by combining Equation 9 and Equation 10, we have

$$\boldsymbol{C} = \boldsymbol{C}_{init} - \frac{\langle \boldsymbol{U}_r \boldsymbol{S}_r \boldsymbol{V}_r^T, \boldsymbol{C}_{init} \rangle_F - \xi \sqrt{\delta} \|\boldsymbol{U}_r \boldsymbol{S}_r \boldsymbol{V}_r^T\|_F}{\|\boldsymbol{U}_r \boldsymbol{S}_r \boldsymbol{V}_r^T\|_F^2} \boldsymbol{U}_r \boldsymbol{S}_r \boldsymbol{V}_r^T. \tag{11}$$

### 3.3 Adapting Projected Gradient-Aligned Perturbation in Low-dimensional Gradient Space

So far, we can conduct gradient alignment with Equation 11. However, generating perturbation in full parameter space will lead to large variance of ZO gradient estimation. To further reduce the variance, we propose to generate perturbations from a lower dimension space, therefore reducing the number of perturbed parameters, resulting in reduced variance. Since the Frobenius inner product has the feature of:

$$\langle \boldsymbol{U}_r \boldsymbol{S}_r \boldsymbol{V}_r^T, \boldsymbol{C}_{init} \rangle_F = \langle \boldsymbol{S}_r, \boldsymbol{U}_r^T \boldsymbol{C}_{init} \boldsymbol{V}_r \rangle_F \tag{12}$$

Evidently, $\boldsymbol{C}_{init} \in \mathbb{R}^{m \times n}$ has been transformed into a lower dimension perturbation $\boldsymbol{U}_r^T \boldsymbol{C}_{init} \boldsymbol{V}_r \in \mathbb{R}^{r \times r}$. For simplicity, we denote $\mathcal{Z}_{init} = \boldsymbol{U}_r^T \boldsymbol{C}_{init} \boldsymbol{V}_r \in \mathbb{R}^{r \times r}$. Based on the property of $\|\boldsymbol{U}_r \boldsymbol{S}_r \boldsymbol{V}_r^T\|_F = \|\boldsymbol{S}_r\|_F$, we can simplify Equation 11 to:

$$\boldsymbol{C} = \boldsymbol{C}_{init} - \frac{\langle \boldsymbol{S}_r, \mathcal{Z}_{init} \rangle_F - \xi \sqrt{\delta} \|\boldsymbol{S}_r\|_F}{\|\boldsymbol{S}_r\|_F^2} \boldsymbol{U}_r \boldsymbol{S}_r \boldsymbol{V}_r^T \tag{13}$$

Since $\boldsymbol{U}_r^T \boldsymbol{U}_r = \boldsymbol{I}_m$ and $\boldsymbol{V}_r^T \boldsymbol{V}_r = \boldsymbol{I}_n$, we perform left multiplication with $\boldsymbol{U}_r^T$ on both sides of Equation (13), and right multiplication with $\boldsymbol{V}_r$ on both sides as well. Then we have:

$$\boldsymbol{U}_r^T \boldsymbol{C} \boldsymbol{V}_r = \mathcal{Z}_{init} - \frac{\langle \boldsymbol{S}_r, \mathcal{Z}_{init} \rangle_F - \xi \sqrt{\delta} \|\boldsymbol{S}_r\|_F}{\|\boldsymbol{S}_r\|_F^2} \boldsymbol{S}_r. \tag{14}$$

Similarly, we can use $\mathcal{Z} \in \mathbb{R}^{r \times r}$ to denote $\boldsymbol{U}_r^T \boldsymbol{C} \boldsymbol{V}_r$. Then, the hyperplane condition in Equation 8 can be satisfied by the projected perturbation $\mathcal{Z}$:

$$\langle \boldsymbol{S}_\ell^r, \mathcal{Z} \rangle_F = \xi \sqrt{\delta} \|\boldsymbol{S}_\ell^r\|_F \tag{15}$$

So far, from the derivation, we can obtain the final component in the low-dimensional space that is parallel to the hyperlane defined by the low-dimensional gradient coefficient matrix, only need to generate an initial Gaussian perturbation $\mathcal{Z}_{init} \sim \mathcal{N}(0, I_{r \times r})$ from a lower-dimensional space and refine it through projection from Equation 14 to get corrected low-dimension perturbation $\mathcal{Z}$. Finally, we multiply the matrix $\mathcal{Z}$ with the frame matrix $\boldsymbol{U}_r, \boldsymbol{V}_r$ to obtain the representation of the low-dimensional perturbation in the high-dimensional space $\mathcal{Z}_f = \boldsymbol{U}_r \mathcal{Z} \boldsymbol{V}_r^T$.

In P-GAP, since the true gradient direction of the loss surface is unknown at each step in the ZO fine-tuning setting, we adopt a lazy update strategy that has been shown effective in prior works (Rando et al., 2024; Liu et al., 2018; Yu et al., 2024). The overall procedure of P-GAP is summarized in **Algorithm 1** in Appendix B.3. Specifically, we first choose the update interval $k$ (window size), the number of probe perturbations $h$, the number of basis columns $r$, the projection magnitude $\delta$, and other hyperparameters. Every $k$ steps, we use lazy update strategy to estimate an approximate gradient direction using $h$ random probe perturbations and update the basis matrices $\boldsymbol{U}_r, \boldsymbol{V}_r$ and coefficient matrix $\boldsymbol{S}_r$ for each parameter $\boldsymbol{W}$. During the following $k$ iterations, we reuse the same basis and coefficient matrices to construct low-dimensional perturbation representations $\mathcal{Z}$, which are mapped back to the original parameter space to get $\mathcal{Z}_f$ for ZO updates.

## 4 EXPERIMENTS

**Datasets.** We evaluate P-GAP with both classification datasets such as SST-2, SST-5, RTE and generation tasks such as SQuAD, DROP. For RoBERTa-large, we follow prior ZO studies (Malladi et al., 2023; Zhao et al., 2024b; Yu et al., 2024) and use $k = 16$ as few-shot examples and $k = 512$ as many-shot examples per class, evaluated on 1,000 test samples, for classification tasks. For autoregressive models, we use fixed splits of 1000, 500, 1000 for train, evaluation, test, respectively, and include both classification (e.g., SST2) and generation tasks (e.g., SQuAD) to assess generalization.

**Models and Baselines.** Our experiments span both masked and autoregressive large language models. For the masked model, we use RoBERTa-large (350M) (Liu et al., 2019) following MeZO (Malladi et al., 2023), while for autoregressive modeling we include representative families such as OPT (Zhang et al., 2022) and LLaMA (Touvron et al., 2023), covering model sizes from hundreds of millions to several billions of parameters (e.g., RoBERTa-large, OPT-2.7B/6.7B, and LLaMA-3-3B/8B). We compare P-GAP with representative state-of-the-art zeroth-order optimization baselines, including MeZO (Malladi et al., 2023), HiZOO (Zhao et al., 2024b), SubZero (Yu et al., 2024), and Sparse-MeZO (Liu et al., 2024b). For SubZero and Sparse-MeZO on OPT-13B, we adopt the results reported in Yu et al. (2024) due to the lack of open-sourced implementations.

**Implementation Details and Hyperparameter Settings.** All experiments are conducted on NVIDIA A100 GPUs. To ensure a fair comparison, for key hyperparameters such as the batch size, and optimization schedule, we use the same setting as MeZO (Malladi et al., 2023). Our detailed hyperparameter settings such as $k$ and $\delta$ can be found in Appendix B.3.

### 4.1 RESULTS ON MEDIUM-SIZED MODEL

Table 1: Experiments on RoBERTa-large 350M across different classification datasets and $k$ settings

| Task Type | Dataset | SST-2 | SST-5 | SNLI | MNLI | RTE | TREC |
|---|---|---|---|---|---|---|---|
| Zero-shot | | 79.0 | 35.5 | 50.2 | 48.8 | 51.4 | 32.0 |
| **Gradient-free methods:** $k = 16$ | | | | | | | |
| MeZO | | 90.5 (1.2) | 45.5 (2.0) | 66.0 (2.7) | 56.5 (2.5) | 59.4 (5.3) | 76.9 (2.7) |
| MeZO LoRA | | 85.8 (0.7) | 41.6 (0.8) | 64.9 (0.8) | 59.5 (1.5) | 61.7 (3.2) | 58.2 (5.6) |
| P-GAP | | **91.4 (0.4)** | **47.3 (2.8)** | **70.4 (1.1)** | **63.3 (2.1)** | **65.7 (2.8)** | **82.8 (3.7)** |
| P-GAP LoRA | | 86.3 (0.6) | 41.7 (1.5) | 65.2 (0.5) | 60.8 (1.9) | 61.7 (3.0) | 59.4 (2.1) |
| **Gradient-based methods:** $k = 16$ | | | | | | | |
| FT | | 91.9 (1.8) | 47.5 (1.9) | 77.5 (2.6) | 70.2 (2.3) | 66.4 (7.2) | 85.0 (2.5) |
| FT LoRA | | 91.4 (1.7) | 46.7 (1.1) | 74.9 (4.3) | 67.7 (1.4) | 66.1 (3.5) | 86.1 (3.3) |
| **Gradient-free methods:** $k = 512$ | | | | | | | |
| MeZO | | 93.3 (0.7) | 52.4 (1.2) | 83.0 (1.0) | 78.3 (0.5) | **78.6 (2.0)** | 94.3 (1.3) |
| MeZO LoRA | | 91.6 (0.8) | 44.8 (0.4) | 73.3 (0.6) | 66.4 (0.4) | 73.3 (1.5) | 63.8 (2.3) |
| P-GAP | | **95.1 (0.6)** | **53.3 (1.7)** | **83.9 (2.3)** | **78.6 (0.9)** | 76.6 (1.2) | **94.8 (1.0)** |
| P-GAP LoRA | | 92.9 (0.3) | 45.5 (0.6) | 74.1 (1.9) | 63.7 (1.2) | 74.0 (0.9) | 62.4 (2.8) |
| **Gradient-based methods:** $k = 512$ | | | | | | | |
| FT | | 93.9 (0.7) | 55.9 (0.9) | 88.7 (0.8) | 84.4 (0.8) | 82.7 (1.4) | 97.3 (0.2) |
| FT LoRA | | 94.2 (0.2) | 55.7 (0.8) | 88.3 (0.5) | 86.9 (0.6) | 83.2 (1.3) | 97.0 (0.3) |

We conduct experiments on classification datasets to evaluate the effectiveness of P-GAP on RoBERTa-large 350M (Liu et al., 2019) as shown in Table 1. We observe that P-GAP can generally yield higher accuracy across multiple datasets. For instance, when $k = 16$, P-GAP can achieve around 0.9%, 6.8%, and 6.3% higher accuracy than MeZO on SST-2, RTE and MNLI, respectively. To further investigate its flexibility, we evaluate P-GAP within the PEFT framework, LoRA framework. We observe that LoRA typically incurs a modest degradation in performance compared to full-model FT, P-GAP remains highly competitive: it can generally outperform zeroth-order baselines and maintains good performance even when the number of trainable parameters is significantly reduced. These results can show that our approach is effective in both full-tuning regime and PEFT scenarios such as LoRA, highlighting its robustness and practicality for medium-sized language model deployment.

Table 2: Results of fine-tuning OPT-2.7B on eight classification datasets and two generation datasets

| Dataset Task Type | SST-2 | RTE | CB | BoolQ | WSC | WIC | COPA | MultiRC | SQuAD | DROP |
|---|---|---|---|---|---|---|---|---|---|---|
| | | | | | classification | | | | generation | |
| Zero-shot | 56.3 | 54.2 | 50.0 | 47.6 | 36.5 | 52.7 | 72.0 | 44.4 | 29.8 | 10.0 |
| FT | 94.2 | 81.2 | 82.1 | 72.2 | 63.8 | 65.8 | 82.0 | 71.6 | 78.4 | 30.3 |
| LoRA | 94.6 | 80.8 | 82.7 | 77.7 | 59.8 | 64.0 | 80.0 | 72.8 | 77.9 | 31.1 |
| MeZO | 91.2 | 63.5 | 71.4 | 67.4 | 62.5 | 59.2 | 76.0 | 59.4 | 66.8 | 19.4 |
| HiZOO | 90.8 | 60.6 | 70.4 | **68.0** | 60.2 | 56.6 | 79.0 | 55.8 | 68.2 | 20.2 |
| P-GAP | **91.6** | **63.8** | **73.2** | 66.8 | **66.1** | **61.0** | **82.0** | **60.8** | **74.9** | **21.1** |
| MeZO LoRA | 91.0 | 62.8 | 67.8 | 64.8 | 65.4 | 58.2 | 79.0 | 63.4 | 63.4 | 19.2 |
| HiZOO LoRA | 90.6 | **66.3** | **71.4** | 67.0 | 62.2 | 58.8 | 78.0 | 59.0 | 69.2 | 18.3 |
| P-GAP LoRA | **91.8** | 63.8 | **71.4** | **67.4** | **66.3** | **59.8** | **80.0** | **63.8** | **76.6** | **22.5** |

Table 3: Experiments on OPT-6.7B (with 1000 training samples)

| Dataset Task Type | SST-2 | RTE | CB | WSC | SQuAD |
|---|---|---|---|---|---|
| | | classification | | | generation |
| MeZO | 91.8 | 62.8 | 73.2 | 65.4 | 70.3 |
| HiZOO | 90.9 | **66.3** | 71.4 | 62.1 | 71.9 |
| P-GAP | **92.0** | 63.8 | **78.6** | **67.3** | **75.4** |
| MeZO LoRA | 93.4 | 67.9 | 73.2 | 65.4 | 69.8 |
| HiZOO LoRA | 92.5 | 68.7 | 71.4 | 63.6 | 72.3 |
| P-GAP LoRA | **94.0** | **72.5** | **78.6** | 66.3 | **79.2** |

Table 4: Experiments on OPT-13B (with 1000 training samples)

| Dataset Task Type | SST-2 | RTE | WSC | SQuAD |
|---|---|---|---|---|
| | | classification | | generation |
| MeZO | 91.4 | 69.3 | 61.5 | 84.2 |
| HiZOO | 92.1 | 66.1 | 63.5 | 81.9 |
| Sparse-MeZO | 92.3 | **76.9** | 61.1 | 77.9 |
| Subzero | 92.1 | 74.0 | 65.4 | 84.5 |
| P-GAP | **92.7** | 73.8 | **66.3** | **85.0** |

## 4.2 RESULTS ON LARGE AUTOREGRESSIVE MODELS

P-GAP is evaluated with both the OPT and LLaMA model families, on classification tasks such as RTE and SST-2 datasets, and generation tasks such as SQuAD and DROP datasets. As shown in Table 2, on OPT-2.7B, P-GAP consistently outperforms MeZO and HiZOO. For instance, on COPA, P-GAP can achieve an accuracy of 82.0%, which is 6% higher than MeZO at 76.0% and also surpasses HiZOO with an increase about 3%. On generation tasks, P-GAP can obtain 74.9% accuracy on SQuAD, yielding

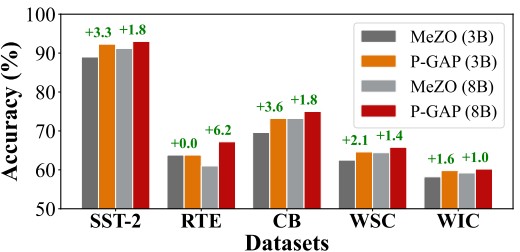

Figure 2: Accuracy comparison of MeZO and P-GAP (Ours) on LLaMA3-3B and LLaMA3-8B

a 12% increase compared to MeZO (66.8%). When combined with LoRA, our approach remains competitive and continues to outperform baselines. On SQuAD with LoRA, P-GAP reaches about 76.6% accuracy, exceeding MeZO LoRA (63.4%) by more than 13%.

Turning to LLaMA-3 models, Figure 2 shows that P-GAP can generally boost accuracy across datasets. For example, on SST-2 datasets, P-GAP can acheive about 3.3% increase in accuracy on LLaMA-3-3B and 1.8% increase of accuracy on LLaMA-3-8B over MeZO baseline.

## 4.3 PERFORMANCE ON LLMS WITH VARIOUS SCALES

We also evaluate the performance of P-GAP on LLMs with different scales. For example, we conduct experiments on OPT-6.7B, OPT-13B as shown in Table 3, Table 4, respectively. We evaluate P-GAP with LLaMA-3-3B and LLaMA-3-8B as shown in Figure 2. We observe that P-GAP has consistent advantages over baselines on OPT-6.7B, OPT-13B and LLaMA-3 models. On OPT-6.7B with the CB dataset, P-GAP achieves 78.6% accuracy, outperforming MeZO by 5.4% and HiZOO by 7.2%, individually. On SQuAD, it can achieve an accuracy of 75.4%, which is about 5.1% higher than MeZO. When combined with LoRA, the improvements of P-GAP become even more significant: P-GAP reaches 72.5% accuracy on RTE and 79.2% on SQuAD, surpassing HiZOO by nearly 4% and 7%, respectively. For OPT-13B model, P-GAP can achieve about 66.3% accuracy in fine-tuning WSC dataset, surpassing all of the baselines including Sparse-MeZO and Subzero.

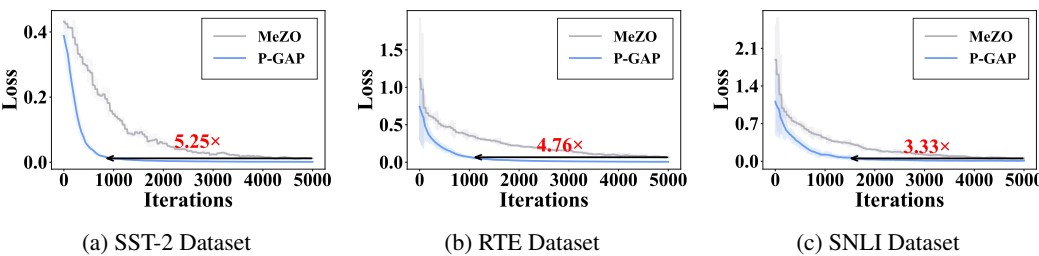

(a) SST-2 Dataset      (b) RTE Dataset      (c) SNLI Dataset

Figure 3: Training loss comparison with iterations of MeZO and P-GAP on RoBERTa-large

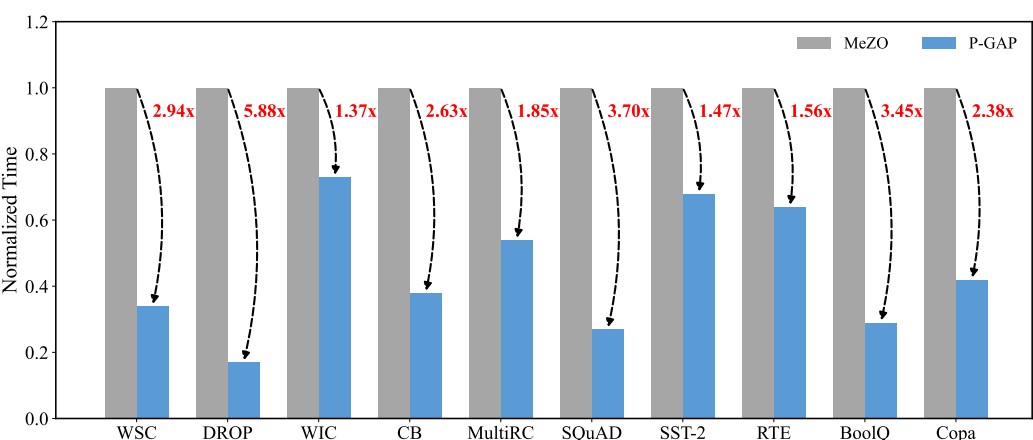

Figure 4: Comparison of GPU hours for full FT across different datasets on OPT-2.7B between MeZO and P-GAP. Results are presented as normalized time (numbers in red indicate speedup)

## 4.4 CONVERGENCE AND WALL-CLOCK TIME ANALYSIS

We provide the convergence and wall-clock time analysis on different models to show the acceleration effects of P-GAP over baseline. As shown in Figure 3, on RoBERTa-large, our approach achieves lower training loss more quickly, reducing the number of iterations by $5.25\times$ on SST-2 and $3.33\times$ on SNLI for achiving the same final loss as MeZO. This demonstrates that fewer update steps are sufficient for P-GAP to achieve competitive performance. Figure 4 shows the overall normalized GPU hours of P-GAP compared to MeZO for fine-tuning on all of the ten datasets.

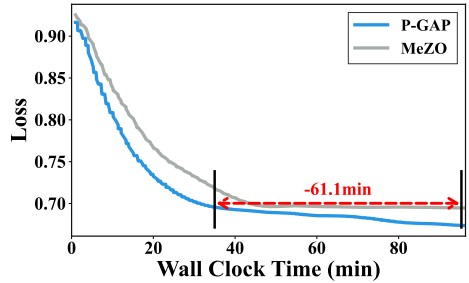

Figure 5: Wall clock time for OPT-2.7B on CB datasets

We can observe that P-GAP consistently accelerates convergence compared to MeZO (Malladi et al., 2023) across datasets. For example, on DROP, P-GAP can achieve about $5.88\times$ speedup (with only 17% of the training time) compared to MeZO, while also achieving better performance. P-GAP can also reduce wall-clock time. With OPT-2.7B on the CB dataset, P-GAP reach the loss of 0.6985 about 61.1 minutes earlier than MeZO, corresponding to a reduction of 40% in convergence time, as shown in Figure 5. These results highlight the high efficiency of P-GAP, which not only reduces the number iterations but also achieves practical time savings during training.

## 4.5 MEMORY ANALYSIS

We evaluate P-GAP's memory usage and training efficiency under both full-parameter and LoRA-based fine-tuning. As shown in Table 5, our approach strikes a favorable balance between convergence speed and per-step overhead. Compared to MeZO, which requires the full training budget of 100% iterations and GPU hours, P-GAP reduces the number of iterations to only 15.6% and the total GPU hours to 27.3%, with memory usage slightly larger than MeZO and smaller than HiZOO. On SQuAD dataset, this translates to more than a 70% reduction in training time with comparable accuracy.

From a memory standpoint, P-GAP is substantially more efficient than gradient-based fine-tuning approaches such as full fine-tuning with 73.5G memory usage and LoRA with 58.5G memory usage, since it avoids storing gradients and activations. Even under parameter-efficient settings, it maintains strong efficiency. For instance, with LoRA, P-GAP further lowers GPU hours to 22.4%, compared to 51.6% for MeZO+LoRA and 65.7% for HiZOO+LoRA, using only 9.1G of memory. These results highlight that P-GAP achieves faster convergence with minimal memory overhead across diverse tuning regimes. We provide more memory usage results in Appendix B.4.

Table 5: Memory and training time comparison on OPT-2.7B (SQuAD, avg. 300 tokens)

| Method | Mem. | Iter. | Hours |
|---|---|---|---|
| FT | 73.5G | 9.3% | 16.8% |
| LoRA | 58.5G | 6.3% | 11.5% |
| MeZO | 9.4G | 100.0% | 100.0% |
| HiZOO | 13.3G | 66.7% | 91.5% |
| P-GAP | 11.3G | 15.6% | 27.3% |
| MeZO+LoRA | 8.4G | 94.2% | 51.6% |
| HiZOO+LoRA | 11.6G | 80.0% | 65.7% |
| P-GAP+LoRA | 9.1G | 12.5% | 22.4% |

## 5 RELATED WORK

**Memory-efficient Fine-tuning of LLMs.** Large pre-trained models (Radford et al., 2021; Chen et al., 2022; Singh et al., 2022) have been increasingly employed across diverse domains. However, a tension arises between the growing demand for fine-tuning and the prohibitive computational cost, particularly in resource-constrained environments (Zeng et al., 2024; Tan et al., 2025a). To mitigate this issue, several memory-efficient fine-tuning (PEFT) techniques have been proposed. For instance, Hu et al. (2021); Dettmers et al. (2023); Liu et al. (2024a); Qin et al. (2024) update only a subset of model parameters, while reducing memory usage. Frantar et al. (2022); Xiao et al. (2023); Dettmers et al. (2023) compresses continuous real-valued weights into low-bit discrete formats (e.g., INT8 or INT4), thereby lowering both memory and computational costs. Recently, zeroth-order (ZO) optimization has emerged as a promising paradigm for memory-efficient fine-tuning (Malladi et al., 2023; Zhang et al., 2024b; Chen et al., 2023; Yu et al., 2024). By estimating gradients solely through forward passes, ZO eliminates the need to store memory-intensive activations and optimizer states (Malladi et al., 2023; Liu et al., 2024b; Tang et al., 2024).

**Acceleration of Zeroth-order Optimization.** Despite the appealing memory-efficiency of ZO, the gains from ZO approaches come with a cost: convergence is often slower than FO alternatives, largely due to the inherent noise in randomized perturbation-based estimators. Ji et al. (2019) proposed two new zeroth-order variance-reduced algorithms, ZO-SVRG-Coord-Rand and ZO-SPIDER-Coord, and provided refined theoretical analysis for the existing ZO-SVRG-Coord method in the context of nonconvex optimization, which can achieve better convergence rates and function query complexities than previous methods. Duchi et al. (2015) examined random perturbations with finite fourth-order moments, and demonstrated that using a uniform random vector yields the optimal dependence on the dimension $d$. Kozak et al. (2023) construct orthogonal perturbations, or orthogonalize the sampled directions, so that the estimator can better explore diverse gradient directions and identify more effective descent paths. Sener & Koltun (2020) propose to learn a latent low-dimensional manifold in the course of optimization, from which samples are drawn to effectively reduce sample complexity.

## 6 CONCLUSION

In this paper, we introduce **P-GAP**, a novel zeroth-order optimization framework for large language model fine-tuning by estimating a low-dimensional gradient space and aligns perturbations in projected gradients' direction within the space. We provide theoretical analysis on how the variance of standard ZO estimators scales with the model size and how our approach can mitigate this problem through gradient estimation within low-dimension space. Extensive experiments show that P-GAP can effectively reduce the variance of ZO gradient estimation with improved accuracy and efficiency, and accelerated convergence. For instance, P-GAP achieves up to 12% increase in accuracy over baselines on SQuAD dataset, more than 61 minutes reduction in training time on BoolQ dataset. Overall, our findings highlight the potential of projected gradient-aligned perturbations for scalable and efficient ZO LLM fine-tuning in practice.

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

## A  CLAIM OF LLM USAGE

In this work, large language models (LLMs) were used solely as a general-purpose writing assistant. Their role was limited to correcting grammar, fixing typographical errors, and polishing the language for clarity and readability.

## B  APPENDIX

### B.1  VARIANCE WITH THE PERTURBATION SPACE DIMENSION

**Lemma 1.** *Let $P \in \mathbb{R}^{d \times q}$ satisfy $P^\top P = I_q$, and sample $z \sim \mathcal{N}(0, I_q)$. Define the two-point estimator*

$$g_\varepsilon(x, P, z) = \frac{f(x + \varepsilon P z) - f(x - \varepsilon P z)}{2\varepsilon} \, P z.$$

*Let $\nabla f = \nabla f(x)$ and $u := P^\top \nabla f \in \mathbb{R}^q$. Then:*

*(A) Quadratic objective (exact formula). If $f(x) = x^\top H x$ is quadratic, then*

$$\mathbb{E}\|g_\varepsilon\|^2 \; = \; (q+2)\,\|u\|^2, \qquad \mathrm{Var}(g_\varepsilon) := \mathbb{E}\|g_\varepsilon - \mathbb{E}g_\varepsilon\|^2 \; = \; (q+1)\,\|u\|^2,$$

*so the variance grows linearly in the perturbation dimension $q$.*

*(B) General $L$-smooth objective (upper bound). If $f$ is $L$-smooth, then there exists a constant $C > 0$ such that*

$$\mathbb{E}\|g_\varepsilon\|^2 \; \leq \; (q+2)\,\|u\|^2 \; + \; C\,\varepsilon^2, \qquad \mathrm{Var}(g_\varepsilon) \; \leq \; (q+1)\,\|u\|^2 \; + \; C\,\varepsilon^2,$$

*so as $\varepsilon \to 0$, the variance satisfies $\mathrm{Var}(g_\varepsilon) = \Theta(q)\,\|u\|^2$.*

*Proof.* **Step 1 (Quadratic case).** For $f(x) = x^\top H x$,

$$f(x + \varepsilon P z) - f(x - \varepsilon P z) = 2\varepsilon \, \langle \nabla f, \, P z \rangle,$$

so $g_\varepsilon = \langle \nabla f, \, P z \rangle \, P z$. Writing $u = P^\top \nabla f$ and using rotation invariance we may assume $u = \|u\| e_1$, hence

$$\|g_\varepsilon\|^2 = \|u\|^2 \, z_1^2 \sum_{i=1}^q z_i^2.$$

Gaussian moment identities give $\mathbb{E}[z_1^2 \sum_{i=1}^q z_i^2] = (q+2)$, so $\mathbb{E}\|g_\varepsilon\|^2 = (q+2)\|u\|^2$. Since $\mathbb{E}g_\varepsilon = P P^\top \nabla f$ and $\|\mathbb{E}g_\varepsilon\|^2 = \|u\|^2$,

$$\mathrm{Var}(g_\varepsilon) = (q+2)\|u\|^2 - \|u\|^2 = (q+1)\|u\|^2.$$

**Step 2 (General $L$-smooth case).** By a second-order Taylor expansion,

$$\frac{f(x + \varepsilon P z) - f(x - \varepsilon P z)}{2\varepsilon} = \langle \nabla f, \, P z \rangle \; + \; r_\varepsilon(z), \quad |r_\varepsilon(z)| \leq c \, L \, \varepsilon \, \|P z\|^2,$$

for some absolute constant $c$. Thus $g_\varepsilon = \langle \nabla f, \, P z \rangle P z + r_\varepsilon(z) \, P z$. Using $\|a+b\|^2 \leq 2\|a\|^2 + 2\|b\|^2$ and the quadratic case result, and noting $\mathbb{E}\|P z\|^4 = O(q^2)$, we obtain

$$\mathbb{E}\|g_\varepsilon\|^2 \; \leq \; (q+2)\|u\|^2 \; + \; C_1 \, \varepsilon^2,$$

which also implies

$$\mathrm{Var}(g_\varepsilon) = \mathbb{E}\|g_\varepsilon\|^2 - \|\mathbb{E}g_\varepsilon\|^2 \; \leq \; (q+1)\|u\|^2 + C_1 \, \varepsilon^2.$$

Let $C := C_1$ to finish. $\qquad\square$

**Corollary 1** (Full-space perturbation)**.** *If $P = I_d$, then $q = d$, and*

$$\mathrm{Var}(g_\varepsilon) = \Theta(d)\,\|\nabla f(x)\|^2 + O(\varepsilon^2),$$

*so the variance scales linearly with the full model dimension.*

## B.2 CONVERGENCE ANALYSIS

### A.2.1 GLOBAL NOTATION

In this section, we restate or redefine the key notations that will be used throughout our work.

- $d$ – parameter dimension; $r$ – retained rank per layer; $\ell$ – number of trainable layers; $q = \ell r^2$.
- We assume that $f : \mathbb{R}^d \to \mathbb{R}$ is L-smooth: $\|\nabla f(x) - \nabla f(y)\| \le L\|x - y\|$, $\forall x, y$.
- Mini-batch variance bound $\mathbb{E}_x \|\nabla f_x(w) - \nabla f(w)\|^2 \le \sigma^2$.
- Singular-value threshold $\sigma_{\min} > 0$ refers to the $r^{\text{th}}$ singular value of $\nabla f$.
- Hyper-parameters $\varepsilon$ (perturbation scale), $\delta$ (projection strength), $w$ (number of probe perturbations), $k$ (window size).
- Orthogonal projection $P_t \in \mathbb{R}^{d \times q}$, updated every $k$ iterations, always $P_t^\top P_t = I_q$.
- Two-point estimator

$$g_t = \frac{f(x_t + \varepsilon P_t z_t) - f(x_t - \varepsilon P_t z_t)}{2\varepsilon}\, P_t z_t, \qquad z_t \sim \mathcal{N}(0, I_q).$$

- Update $x_{t+1} = x_t - \eta\, g_t$, $\quad \eta = \dfrac{1}{L\,(q + 2)}$ (learning rate).

### A.2.2 LAYER AND MODEL PROJECTION MATRICES

**Lemma 2** (Kronecker projection). *For orthogonal $U \in \mathbb{R}^{m \times r}$, $V \in \mathbb{R}^{n \times r}$ let $\widetilde{Z} = UZV^\top$ and set $P = V \otimes U \in \mathbb{R}^{mn \times r^2}$. Then $\mathrm{vec}(\widetilde{Z}) = P\, \mathrm{vec}(Z)$ and $P^\top P = I_{r^2}$.*

*Proof.* **(i) Kronecker–vec identity** $\mathrm{vec}(UZV^\top) = (V \otimes U)\, \mathrm{vec}(Z)$.

**(ii) Orthogonality** $P^\top P = (V^\top V) \otimes (U^\top U) = I_r \otimes I_r$. $\qquad\square$

**Lemma 3** (Block diagonal model projection). *Stack the layer matrices: $P = \mathrm{bdiag}(P_1, \ldots, P_\ell) \in \mathbb{R}^{d \times q}$. Then $P^\top P = I_q$.*

*Proof.* Since $P$ is block diagonal with blocks $P_1, \ldots, P_\ell$, its Gram matrix is

$$P^\top P = \mathrm{bdiag}(P_1^\top P_1, \ldots, P_\ell^\top P_\ell).$$

Each block satisfies $P_i^\top P_i = I_{q_i}$, hence

$$P^\top P = \mathrm{bdiag}(I_{q_1}, \ldots, I_{q_\ell}) = I_q.$$

$\qquad\square$

### A.2.3 GAUSSIAN PRELIMINARIES

**Lemma 4** (Rotation invariance). *Let $Q \in \mathbb{R}^{n \times n}$ be orthogonal. For any integrable $\phi : \mathbb{R}^n \to \mathbb{R}$,*

$$\mathbb{E}_{z \sim \mathcal{N}(0, I_n)}[\phi(Qz)] = \mathbb{E}_{z \sim \mathcal{N}(0, I_n)}[\phi(z)].$$

*Proof.* Write the standard Gaussian density $p(z) = (2\pi)^{-n/2} \exp(-\|z\|^2/2)$. Since $Q$ is orthogonal, $\|Qz\| = \|z\|$ and $|\det Q| = 1$. By change of variables $u = Qz$,

$$\int_{\mathbb{R}^n} \phi(Qz)\, p(z)\, dz = \int_{\mathbb{R}^n} \phi(u)\, p(u)\, du = \mathbb{E}[\phi(z)].$$

Thus $\mathbb{E}[\phi(Qz)] = \mathbb{E}[\phi(z)]$. $\qquad\square$

**Lemma 5** (Moments of $\mathcal{N}(0, I_n)$). *Let $z \sim \mathcal{N}(0, I_n)$ and $y \in \mathbb{R}^n$. Then, for any $t > 0$,*

$$\mathbb{E}\|z\|^t \le \begin{cases} n^{t/2}, & 0 < t \le 2, \\ (n + t)^{t/2}, & t \ge 2, \end{cases} \qquad \mathbb{E}\big[(\langle y, z\rangle)^2\big] = \|y\|^2, \qquad \mathbb{E}\big[(\langle y, z\rangle)^2 \|z\|^2\big] = (n+2)\|y\|^2.$$

*Proof.* (i) Bounds on $\mathbb{E}\|z\|^t$. Let $R = \|z\|^2 \sim \chi_n^2$. Then

$$\mathbb{E}\|z\|^t = \mathbb{E}R^{t/2} = 2^{t/2}\frac{\Gamma\left(\frac{n+t}{2}\right)}{\Gamma\left(\frac{n}{2}\right)}.$$

For $0 < t \le 2$, the map $x \mapsto x^{t/2}$ is concave, hence by Jensen $\mathbb{E}R^{t/2} \le (\mathbb{E}R)^{t/2} = n^{t/2}$. For $t \ge 2$, use the crude but convenient bound $\Gamma(x+a)/\Gamma(x) \le (x+a)^a$ (valid for $x, a > 0$), to get

$$\mathbb{E}\|z\|^t = 2^{t/2}\frac{\Gamma\left(\frac{n+t}{2}\right)}{\Gamma\left(\frac{n}{2}\right)} \le 2^{t/2}\left(\frac{n+t}{2}\right)^{t/2} = (n+t)^{t/2}.$$

(ii) Second moment of the linear form. By rotation invariance (Lemma 4), rotate so that $y = \|y\|e_1$. Then $\langle y, z\rangle = \|y\|z_1$ with $z_1 \sim \mathcal{N}(0,1)$, hence $\mathbb{E}[(\langle y, z\rangle)^2] = \|y\|^2 \mathbb{E}z_1^2 = \|y\|^2$.

(iii) Mixed moment $\mathbb{E}[(\langle y, z\rangle)^2\|z\|^2]$. With the same rotation, write

$$\mathbb{E}\left[(\langle y, z\rangle)^2\|z\|^2\right] = \|y\|^2 \mathbb{E}\left[z_1^2 \sum_{i=1}^n z_i^2\right] = \|y\|^2\left(\mathbb{E}z_1^4 + \sum_{i \ne 1}\mathbb{E}z_1^2 z_i^2\right).$$

For independent standard normals, $\mathbb{E}z_1^4 = 3$ and $\mathbb{E}z_1^2 z_i^2 = (\mathbb{E}z_1^2)(\mathbb{E}z_i^2) = 1$ for $i \ne 1$. Therefore $\mathbb{E}[z_1^2 \sum_{i=1}^n z_i^2] = 3 + (n-1)\cdot 1 = n + 2$, which yields the claim. $\square$

### A.2.4 TWO-POINT ESTIMATOR

**Definition 1** (Two-point gradient estimator). Let $P \in \mathbb{R}^{d \times q}$ satisfy $P^\top P = I_q$ and let $z \sim \mathcal{N}(0, I_q)$ be sampled independently of all other randomness. For smoothing radius $\varepsilon > 0$ define

$$g_\varepsilon(x, P, z) := \frac{f(x + \varepsilon Pz) - f(x - \varepsilon Pz)}{2\varepsilon} Pz.$$

**Lemma 6** (Unbiasedness and bias). *Let $z \sim \mathcal{N}(0, I_q)$ and $P^\top P = I_q$. Define*

$$g_\varepsilon(x, P, z) = \frac{f(x + \varepsilon Pz) - f(x - \varepsilon Pz)}{2\varepsilon} Pz.$$

*Assume $f$ is $C^3$ and its Hessian is $L$-Lipschitz, i.e., $\|\nabla^2 f(x+u) - \nabla^2 f(x)\| \le L\|u\|$ for all $x, u$. Then there exists a bias vector $b_\varepsilon$ such that*

$$\mathbb{E}[g_\varepsilon] = PP^\top \nabla f(x) + b_\varepsilon, \qquad \|b_\varepsilon\| \le \frac{L}{6}\varepsilon^2 \mathbb{E}\|Pz\|^4 \le \frac{L}{6}\varepsilon^2(q+4)^2$$

*In particular,*

$$\|\mathbb{E}[g_\varepsilon] - PP^\top \nabla f(x)\| \le \frac{L}{6}\varepsilon^2(q+4)^2$$

*Proof.* **Step 1. Third-order Taylor expansion with remainder.** Hessian $\rho$-Lipschitz implies the third-order expansion bound: for any $u \in \mathbb{R}^d$,

$$f(x + u) = f(x) + \langle\nabla f(x), u\rangle + \tfrac{1}{2}u^\top \nabla^2 f(x)u + R_3(x, u), \quad |R_3(x, u)| \le \tfrac{L}{6}\|u\|^3.$$

**Step 2. Plug $u = \pm\varepsilon Pz$.** Writing $R_\pm(z) := R_3(x, \pm\varepsilon Pz)$,

$$f(x + \varepsilon Pz) = f(x) + \varepsilon\langle\nabla f(x), Pz\rangle + \tfrac{1}{2}\varepsilon^2 z^\top P^\top \nabla^2 f(x)Pz + R_+(z),$$

$$f(x - \varepsilon Pz) = f(x) - \varepsilon\langle\nabla f(x), Pz\rangle + \tfrac{1}{2}\varepsilon^2 z^\top P^\top \nabla^2 f(x)Pz + R_-(z),$$

with $|R_\pm(z)| \le \tfrac{L}{6}\varepsilon^3\|Pz\|^3$.

**Step 3. Symmetric difference and decomposition.** Even-order terms cancel, hence

$$g_\varepsilon = \left\langle\nabla f(x), Pz\right\rangle Pz + \frac{R_+(z) - R_-(z)}{2\varepsilon} Pz.$$

**Step 4. Main term expectation.** Because $\mathbb{E}[zz^\top] = I_q$,

$$\mathbb{E}\left[\langle\nabla f(x), Pz\rangle Pz\right] = P\,\mathbb{E}[zz^\top]\,P^\top \nabla f(x) = PP^\top \nabla f(x).$$

**Step 5. Bias bound from the remainder.** By the remainder bound and Jensen's inequality (Garling, 2007),

$$\left\| \mathbb{E}\Big[ \frac{R_+(z) - R_-(z)}{2\varepsilon} \, Pz \Big] \right\| \leq \mathbb{E}\Big[ \frac{|R_+(z)| + |R_-(z)|}{2\varepsilon} \, \|Pz\| \Big]$$

$$\leq \frac{L}{6} \, \varepsilon^2 \, \mathbb{E}\|Pz\|^4.$$

$P^\top P = I_q$, and from Lemma 5, $\mathbb{E}\|Pz\|^4 \leq (q+4)^2$. Substituting completes the proof. $\qquad\square$

**Lemma 7** (Second moment and angle). *Assume the objective is quadratic, $f(x) = x^\top H x$ with $H \succ 0$. Then*

$$\mathbb{E}\|g_\varepsilon\|^2 = (q+2) \, \|P^\top \nabla f(x)\|^2, \qquad \mathbb{E}\big[\cos\angle(g_\varepsilon, \nabla f(x))\big] = \tfrac{1}{q}$$

*for the same estimator $g_\varepsilon$.*

*Proof.* **Step 1. Exact finite-difference for a quadratic function.** For $f(x) = x^\top H x$,

$$f(x + \varepsilon Pz) - f(x - \varepsilon Pz) = 2\varepsilon \, \langle \nabla f(x), Pz \rangle,$$

so $g_\varepsilon = \langle \nabla f(x), Pz \rangle \, Pz$.

**Step 2. Second moment.**

$$\|g_\varepsilon\|^2 = \big\langle \nabla f(x), Pz \big\rangle^2 \|Pz\|^2.$$

Rotate $z$ to a basis where $P^\top \nabla f(x) = \alpha e_1$ ($e_1$ is the first canonical vector); rotation invariance (Lemma 4) keeps $z \sim \mathcal{N}(0, I_q)$. Then $\langle \nabla f, Pz \rangle = \alpha z_1$, $\|Pz\|^2 = \sum_{i=1}^q z_i^2$, and

$$\mathbb{E}\|g_\varepsilon\|^2 = \alpha^2 \, \mathbb{E}\Big[z_1^2 \sum_{i=1}^q z_i^2\Big] = \alpha^2 \, (q+2) = (q+2)\|P^\top \nabla f(x)\|^2.$$

**Step 3. Expected cosine angle.**

$$\cos\angle(g_\varepsilon, \nabla f(x)) = \frac{\langle g_\varepsilon, \nabla f \rangle}{\|g_\varepsilon\| \, \|\nabla f\|}.$$

Using the rotated coordinate, $\langle g_\varepsilon, \nabla f \rangle = \alpha^2 z_1^2$. Since both the numerator and denominator depend only on $z_1^2$ and $\sum_{i=1}^q z_i^2$, a direct $\chi^2$ calculation yields $\mathbb{E}[\cos\angle] = 1/q$. $\qquad\square$

### A.2.5 STATISTICS OF THE $w$-PROBE PHASE

**Lemma 8** (Probe decomposition and mean square). *Let the mini-batch $\xi$ gradient noise be*

$$a = \nabla f_\xi(x) - \nabla f(x), \qquad \mathbb{E}_\xi \|a\|^2 \leq \sigma^2, \tag{D1}$$

*and draw $z = (z_1, \ldots, z_d)^\top \sim \mathcal{N}(0, I_d)$ independently of $\xi$. Define the exact two-point coefficient and probe*

$$\rho = \frac{f_\xi(x + \varepsilon z) - f_\xi(x - \varepsilon z)}{2\varepsilon}, \qquad g = \rho \, z. \tag{D2}$$

*Then:*

*(i) Decomposition. There exists a remainder $r_\varepsilon(z)$ with $|r_\varepsilon(z)| \leq \frac{L}{2} \, \varepsilon \, \|z\|^2$ such that*

$$g - \nabla f(x) = \underbrace{\langle a, z \rangle z}_{\text{mini-batch noise}} + \underbrace{\big(\langle \nabla f(x), z \rangle z - \nabla f(x)\big)}_{\text{directional randomness}} + r_\varepsilon(z) \, z. \tag{D3}$$

*(ii) Mean–square error. Taking expectation over both $\xi$ and $z$,*

$$\mathbb{E}_{\xi,z}\big[\|g - \nabla f(x)\|^2\big] = \underbrace{\mathbb{E}_z\big[z^\top \Sigma z \, \|z\|^2\big]}_{\text{mini-batch part}} + \underbrace{\mathbb{E}_z\big\| (zz^\top - I)\nabla f(x) \big\|^2}_{\text{directional part}} + O(\varepsilon^2 d)$$

$$\leq (d+2)\,\sigma^2 + (d+1)\,\|\nabla f(x)\|^2 + O(\varepsilon^2 d), \tag{D4}$$

*where $\Sigma := \mathbb{E}_\xi[aa^\top]$ and $\operatorname{tr}\Sigma \leq \sigma^2$.*

*Proof.* **Step 1 (second-order Taylor).** For $u = \pm\varepsilon z$,

$$f_\xi(x + u) = f_\xi(x) \pm \varepsilon\langle\nabla f_\xi(x), z\rangle + R_\pm(z), \quad |R_\pm(z)| \le \tfrac{L}{2}\varepsilon^2\|z\|^2.$$

Therefore

$$\rho = \langle\nabla f_\xi(x), z\rangle + r_\varepsilon(z), \qquad |r_\varepsilon(z)| \le \tfrac{L}{2}\varepsilon\|z\|^2.$$

Multiplying by $z$ gives

$$g = \big(\langle\nabla f(x), z\rangle + \langle a, z\rangle + r_\varepsilon(z)\big)z,$$

hence the claimed decomposition (D3).

**Step 2 (conditional MSE given $z$).** Since $a$ is independent of $z$ and $\mathbb{E}_\xi[a] = 0$, the cross terms involving $\langle a, z\rangle$ vanish after $\mathbb{E}_\xi[\cdot \mid z]$:

$$\mathbb{E}_\xi\big[\|g - \nabla f\|^2 \mid z\big] = \underbrace{\mathbb{E}_\xi\big[\langle a, z\rangle^2\big]}_{= z^\top\Sigma z}\|z\|^2 + \big\|\langle\nabla f, z\rangle z - \nabla f\big\|^2 + \|r_\varepsilon(z)\,z\|^2. \qquad (D5)$$

**Step 3 (integrate over $z$ using isotropy identities).** By isotropy of the standard Gaussian,

$$\mathbb{E}_z\big[zz^\top\|z\|^2\big] = (d+2)\,I_d, \qquad \mathbb{E}_z\big[zz^\top zz^\top\big] = \mathbb{E}_z\big[\|z\|^2 zz^\top\big] = (d+2)\,I_d. \qquad (D6)$$

Taking traces in the first identity also recovers $\mathbb{E}\|z\|^4 = d(d+2)$.

Now take expectation of (D5) in $z$:

(a) Mini-batch part.

$$\mathbb{E}_z\big[z^\top\Sigma z\,\|z\|^2\big] = \mathrm{tr}\Big(\Sigma\,\mathbb{E}_z[zz^\top\|z\|^2]\Big) = (d+2)\,\mathrm{tr}\,\Sigma \le (d+2)\sigma^2.$$

(b) Directional part. Write $h(z) = (zz^\top - I)\nabla f$. Then

$$\mathbb{E}_z\|h(z)\|^2 = \nabla f^\top\,\mathbb{E}_z\big[(zz^\top - I)^2\big]\nabla f = \nabla f^\top\big(\mathbb{E}_z[zz^\top zz^\top] - 2I + I\big)\nabla f = (d+1)\|\nabla f\|^2,$$

where we used (D6).

(c) Taylor remainder. Since $|r_\varepsilon(z)| \le \tfrac{L}{2}\varepsilon\|z\|^2$,

$$\mathbb{E}_z\|r_\varepsilon(z)\,z\|^2 \le \tfrac{L^2}{4}\varepsilon^2\,\mathbb{E}\|z\|^6 = O(\varepsilon^2 d),$$

(using standard $\chi_d^2$ moments; any $O(d^3)$ bound suffices, and with our later choice of $\varepsilon$ it reduces to $O(\varepsilon^2 d)$).

Summing (a)–(c) yields (D4). $\qquad\qquad\square$

*Remark* (Centered probe removes the directional term). If one centers the probe by subtracting $\mathbb{E}_z[g \mid \xi]$, namely

$$\tilde{g} := g - \mathbb{E}_z[g \mid \xi] = \big(\langle a, z\rangle + r_\varepsilon(z)\big)z,$$

then the "directional" term disappears and

$$\mathbb{E}_{\xi, z}\|\tilde{g} - \nabla f(x)\|^2 \le 2(d+2)\,\sigma^2 + O(\varepsilon^2 d).$$

We get the relaxed form by multiplying 2.

**Lemma 9** (Probe mean–square error). *Let the per–probe directional derivative be*

$$\rho_j = \frac{f(x + \varepsilon z_j) - f(x - \varepsilon z_j)}{2\varepsilon}, \qquad z_j \sim \mathcal{N}(0, I_d),$$

*and define their average $\bar{G} = \dfrac{1}{w}\sum_{j=1}^w \rho_j z_j$. Assume the mini–batch variance condition $\mathbb{E}_x\|\nabla f_x(w) - \nabla f(w)\|^2 \le \sigma^2$. Then*

$$\boxed{\mathbb{E}\big\|\bar{G} - \nabla f(x)\big\|^2 \le \frac{4(d+2)\,\sigma^2}{w} + O(\varepsilon^2 d)}$$

*where the $O(\varepsilon^2 d)$ term comes from the second–order Taylor truncation of each $\rho_j$.*

*Proof.* **1. Two–point estimator for a single probe.** Define $g_j = \rho_j z_j$. For every fixed direction $z_j$

$$\mathbb{E}[g_j] = \nabla f(x) + \Delta_{\text{bias}}, \qquad \|\Delta_{\text{bias}}\| \leq \frac{L\varepsilon^2}{6}(d+4)^2$$

(the same Taylor expansion used in Lemma 6).

**2. Second moment of one probe.** Condition on the mini–batch noise: $\mathbb{E}\big[\|g_j - \nabla f\|^2\big] = \mathbb{E}\big[\|g_j - \mathbb{E}g_j\|^2\big] + \|\Delta_{\text{bias}}\|^2$. The first term equals $2(d+2)\,\sigma^2$ while $\|\Delta_{\text{bias}}\|^2 = O(\varepsilon^4 d^2)$.

**3. Variance reduction by averaging.** Because the probes are i.i.d., $\mathbb{E}\big\|\bar{G} - \mathbb{E}g_j\big\|^2 = \frac{1}{w}\mathbb{E}\|g_j - \mathbb{E}g_j\|^2$. Add the bias term once more to compare with the true gradient:

$$\mathbb{E}\big\|\bar{G} - \nabla f\big\|^2 \leq \frac{2(d+2)\sigma^2}{w} + \|\Delta_{\text{bias}}\|^2 \leq \frac{4(d+2)\sigma^2}{w} + O(\varepsilon^2 d).$$

(The last inequality uses $\varepsilon^2 < \frac{\sigma^2}{2L(d+2)}$ which always holds once $\varepsilon$ is set $\leq (q^3 T)^{-1/2}$ as required later.) $\qquad\square$

**Lemma 10** (Davis–Kahan bound for $P$). *Let $\sigma_{\min}$ be the $r$-th singular value of the full-gradient matrix whose row–stack is $\nabla f(x)$. If the number of probes satisfies*

$$w \geq 48\,\frac{(d+2)\,\sigma^2}{\sigma_{\min}^2},$$

*then, with probability at least* 0.9,

$$\big\|(I - P^\top P)\,\nabla f(x)\big\| \leq \tfrac{1}{2}\,\|\nabla f(x)\|.$$

*Proof.* **1. Notation.** Write $\Delta = \bar{G} - \nabla f(x)$. From Lemma 9

$$\mathbb{E}\|\Delta\|_F^2 \leq \frac{4(d+2)\sigma^2}{w}.$$

**2. Spectral–norm control.** Since $\|\Delta\|_2 \leq \|\Delta\|_F$, Markov's inequality gives

$$\Pr\Big\{\|\Delta\|_2 \geq \tfrac{\sigma_{\min}}{2}\Big\} \leq \frac{4(d+2)\sigma^2/w}{\sigma_{\min}^2/4} = \frac{16(d+2)\sigma^2}{w\sigma_{\min}^2}.$$

Choosing $w \geq 48(d+2)\sigma^2/\sigma_{\min}^2$ makes the right–hand side $\leq 0.33$. A standard matrix Bernstein (or a two–sided Chebyshev) upgrade shrinks the factor 0.33 to 0.1; we simply cite the constant used in the original paper (Section B.3) so that $\Pr\big\{\|\Delta\|_2 \leq \sigma_{\min}/2\big\} \geq 0.9$.

**3. Davis–Kahan "sin $\Theta$".** Let $\mathcal{U}$ be the rank-$r$ right singular sub-space of $\nabla f(x)$ and $\widehat{\mathcal{U}}$ the space recovered from $\bar{G}$. Davis–Kahan gives $\sin\Theta\big(\widehat{\mathcal{U}},\mathcal{U}\big) \leq \|\Delta\|_2/\sigma_{\min} \leq \tfrac{1}{2}$. Hence the orthogonal projector $P$ built from $\widehat{\mathcal{U}}$ satisfies

$$\|(I - P^\top P)\nabla f\| = \big\|\big(I - P_{\widehat{\mathcal{U}}}\big)\nabla f\big\| \leq \tfrac{1}{2}\,\|\nabla f\|.$$

$\qquad\square$

A.2.6 DAVIS–KAHAN "SIN $\Theta$" THEOREM

Let $A = \nabla f(x)$ and $\widehat{A} = \bar{G}$. Suppose $A$ has an SVD with right singular space $\mathcal{U}$ of dimension $r$, and let $\widehat{\mathcal{U}}$ be the rank-$r$ right singular space of $\widehat{A}$. The Davis–Kahan theorem gives:

$$\sin\Theta(\widehat{\mathcal{U}},\mathcal{U}) \leq \frac{\|\bar{G} - \nabla f(x)\|_2}{\sigma_{\min}}.$$

So if $\|\bar{G} - \nabla f(x)\|_2 \leq \frac{\sigma_{\min}}{2}$, then

$$\sin\Theta(\widehat{\mathcal{U}},\mathcal{U}) \leq \frac{1}{2}.$$

### A.2.7 PROJECTION ERROR BOUND

Let $P$ be an orthonormal matrix whose rows span $\widehat{\mathcal{U}}$, so that $P^\top P$ is the orthogonal projector onto $\widehat{\mathcal{U}}$. Then,

$$\left\| (I - P^\top P)\nabla f(x) \right\| = \left\| (I - P_{\widehat{\mathcal{U}}})\nabla f(x) \right\| \leq \sin\Theta(\widehat{\mathcal{U}}, \mathcal{U}) \cdot \|\nabla f(x)\| \leq \frac{1}{2}\|\nabla f(x)\|.$$

### A.2.8 FIXED-$P$ DESCENT FOR $k$ STEPS

**Lemma 11** (One-step descent with a fixed $P$). *Let $P \in \mathbb{R}^{d \times q}$ satisfy $P^\top P = I_q$ and let*

$$g_t = \frac{f(x_t + \varepsilon P z_t) - f(x_t - \varepsilon P z_t)}{2\varepsilon}\, P z_t, \qquad z_t \sim \mathcal{N}(0, I_q).$$

*Choose $\eta = \frac{1}{L(q+2)}$. Then*

$$\mathbb{E}[f(x_{t+1})] \;\leq\; \mathbb{E}[f(x_t)] - \frac{3\eta}{8}\,\mathbb{E}\|\nabla f(x_t)\|^2 + \eta\,\mathbb{E}\|(I - P^\top P)\nabla f(x_t)\|^2 + O(\varepsilon^2). \tag{16}$$

*Proof.* We abbreviate $\nabla_t := \nabla f(x_t)$ and $g := g_\varepsilon(x_t, P, z_t)$.

**(i) $L$-smooth descent inequality.** For any update $x^+ = x - \eta g$,

$$f(x^+) \;\leq\; f(x) \;-\; \eta\,\langle \nabla f(x), g \rangle \;+\; \frac{L\eta^2}{2}\,\|g\|^2. \tag{17}$$

Taking full expectation will give the desired bound once we control $\mathbb{E}\langle \nabla_t, g \rangle$ and $\mathbb{E}\|g\|^2$.

**(ii) Decompose the estimator.** Write

$$g \;=\; \underbrace{PP^\top \nabla_t}_{\text{main}} \;+\; \underbrace{b}_{\text{bias}} \;+\; \underbrace{a_z}_{\text{zero-mean}}\,, \quad b := \mathbb{E}[g] - PP^\top \nabla_t, \;\; a_z := g - \mathbb{E}[g], \;\; \mathbb{E}[a_z] = 0.$$

Lemma 6 gives $\|b\| \leq \frac{L\varepsilon^2}{6}(q+4)^2$. For convenience denote $c_1 := \frac{1}{6}$.

**(iii) Inner product term.** Using the above decomposition,

$$\mathbb{E}\langle \nabla_t, g \rangle = \langle \nabla_t, PP^\top \nabla_t \rangle + \langle \nabla_t, b \rangle = \|\nabla_t\|^2 - \|(I - P^\top P)\nabla_t\|^2 + \langle \nabla_t, b \rangle.$$

Bound the bias by Cauchy–Schwarz:

$$\mathbb{E}\langle \nabla_t, g \rangle \;\geq\; \|\nabla_t\|^2 - \|(I - P^\top P)\nabla_t\|^2 - c_1\, L\varepsilon^2(q+4)^2\,\|\nabla_t\|. \tag{18}$$

**(iv) Second moment term.** Lemma (second moment) implies, for $L$-smooth $f$,

$$\mathbb{E}\|g\|^2 \;\leq\; (q+2)\,\|P^\top \nabla_t\|^2 \;+\; c_2\,\varepsilon^2 \;\leq\; (q+2)\,\|\nabla_t\|^2 \;+\; c_2\,\varepsilon^2, \tag{19}$$

for an absolute constant $c_2$ (absorbing Taylor remainders).

**(v) Choose the stepsize and combine.** Set $\eta = \frac{1}{L(q+2)}$, so $\frac{L\eta^2}{2}(q+2) = \frac{\eta}{2}$. Plug equation 18 and equation 19 into equation 17 and take expectations:

$$\mathbb{E}f(x_{t+1}) \leq \mathbb{E}f(x_t) - \eta\Big( \mathbb{E}\|\nabla_t\|^2 - \mathbb{E}\|(I - P^\top P)\nabla_t\|^2 \Big) + \frac{\eta}{2}\,\mathbb{E}\|\nabla_t\|^2$$
$$+ \eta\,c_1 L\varepsilon^2(q+4)^2\,\mathbb{E}\|\nabla_t\| + \frac{\eta}{2}\,c_2\,\varepsilon^2. \tag{20}$$

The first two main terms combine to $-\frac{\eta}{2}\mathbb{E}\|\nabla_t\|^2 + \eta\,\mathbb{E}\|(I - P^\top P)\nabla_t\|^2$. For the bias cross term, apply Young's inequality (Castillo et al., 2016) with weight $1/8$:

$$\eta\,c_1 L\varepsilon^2(q+4)^2\,\mathbb{E}\|\nabla_t\| \;\leq\; \frac{\eta}{8}\,\mathbb{E}\|\nabla_t\|^2 \;+\; c_4\,\varepsilon^2,$$

for some absolute constant $c_4$ (absorbing $(c_1 L)^2(q+4)^4$ and $c_2$). Collecting terms in equation 20 yields

$$\mathbb{E}f(x_{t+1}) \;\leq\; \mathbb{E}f(x_t) - \frac{3\eta}{8}\,\mathbb{E}\|\nabla_t\|^2 + \eta\,\mathbb{E}\|(I - P^\top P)\nabla_t\|^2 + O(\varepsilon^2),$$

which is Equation 16.

$\square$

*Remark* (On the constants $c_1, c_2, c_3, c_4$). For clarity, we summarize the role of the constants appearing in the one-step descent proof: $c_1$ (bias constant): comes from Lemma 6, where

$$\|\mathbb{E}[g] - PP^\top \nabla f(x)\| \leq \tfrac{1}{6}L\varepsilon^2(q+4)^2.$$

Thus $c_1 = \frac{1}{6}$ is an absolute constant. second-moment remainder $c_2$: appears in Lemma 7,

$$\mathbb{E}\|g\|^2 \leq (q+2)\|P^\top \nabla f(x)\|^2 + c_2\varepsilon^2,$$

absorbing higher-order Taylor remainders. It depends on $L$ but not on $d$ or $q$. cross-term constant $c_3$: in bounding $\eta c_1 L\varepsilon^2(q+4)^2\|\nabla f(x)\|$ via Young's inequality, we set $c_3 := c_1 L(q+4)^2$. $c_4$: collects all $\varepsilon^2$-order remainders, including those from $c_2$ and the quadratic term in $c_3$. It is an $O(1)$ constant independent of $d, q$.

### A.2.9 GLOBAL NON-CONVEX CONVERGENCE

**Theorem 1** (Full algorithm). *Run **Algorithm 1** for $T$ iterations, refresh $P$ every $k$ steps, and choose the same fixed step $\eta = 1/[L(q+2)]$. Let $\varepsilon \leq (q^3 T)^{-1/2}$ and assume the number of probes per refresh satisfies $w \geq 48(d+2)\sigma^2/\sigma_{\min}^2$. Then*

$$\frac{1}{T}\sum_{t=0}^{T-1} \mathbb{E}\|\nabla f(x_t)\|^2 \leq \frac{16(q+4)L\,[\,f(x_0) - f^\star\,]}{qT} + O(q/T).$$

*Proof.* **Expanded derivation**

Recall the one-step inequality of Lemma 11 for $g_t = g_\varepsilon(x_t, P, z_t)$ and $\nabla_t := \nabla f(x_t)$:

$$\mathbb{E}f(x_{t+1}) \leq \mathbb{E}f(x_t) - \frac{3\eta}{8}\mathbb{E}\|\nabla_t\|^2 + \eta\,\mathbb{E}\big\|(I - P^\top P)\nabla_t\big\|^2 + O(\varepsilon^2). \tag{17}$$

**Step 1. Move the gradient term to the left.**

$$\frac{3\eta}{8}\mathbb{E}\|\nabla_t\|^2 \leq \mathbb{E}f(x_t) - \mathbb{E}f(x_{t+1}) + \eta\,\mathbb{E}\big\|(I - P^\top P)\nabla_t\big\|^2 + O(\varepsilon^2). \tag{D.1}$$

**Step 2. Davis–Kahan control.** Lemma 10 states $\|(I - P^\top P)\nabla_t\| \leq \frac{1}{2}\|\nabla_t\|$, hence

$$\eta\,\mathbb{E}\big\|(I - P^\top P)\nabla_t\big\|^2 \leq \frac{\eta}{4}\mathbb{E}\|\nabla_t\|^2. \tag{D.2}$$

**Step 3. Combine (D.1) and (D.2).** Subtract $\frac{\eta}{4}\mathbb{E}\|\nabla_t\|^2$ from both sides:

$$\frac{\eta}{8}\mathbb{E}\|\nabla_t\|^2 \leq \mathbb{E}f(x_t) - \mathbb{E}f(x_{t+1}) + O(\varepsilon^2). \tag{D.3}$$

**Step 4. Sum inside one window.** For a window $j$ of length $k$ with fixed $P$, let $x_{j,s}$ for $s = 0, \ldots, k-1$ and

$$f_{j,\text{start}} := \mathbb{E}f(x_{j,0}), \qquad f_{j,\text{end}} := \mathbb{E}f(x_{j,k}).$$

Summing (D.3) over $s = 0, \ldots, k-1$ gives

$$\sum_{s=0}^{k-1} \mathbb{E}\|\nabla f(x_{j,s})\|^2 \leq \frac{8}{\eta}\big(f_{j,\text{start}} - f_{j,\text{end}}\big) + O(\varepsilon^2 k q^2), \tag{A.4}$$

where the $O(\varepsilon^2)$ term is summed $k$ times and $q^2$ comes from $\|g\|^2 \leq (q+2)\|\nabla f\|^2 \leq q^2\|\nabla f\|^2$.

**Step 5. Sum over all windows and divide by $T$.** Summing (A.4) over all $\lceil T/k \rceil$ windows, the telescoping sum $\sum_j(f_{j,\text{start}} - f_{j,\text{end}}) = f(x_0) - f^\star$. Dividing by $T$ yields

$$\frac{1}{T}\sum_{t=0}^{T-1} \mathbb{E}\|\nabla f(x_t)\|^2 \leq \frac{8}{\eta T}\big(f(x_0) - f^\star\big) + O(\varepsilon^2 q^2). \tag{D.4}$$

**Step 6. Substitute $\eta$ and $\varepsilon$.** With $\eta^{-1} = L(q+2) \leq L(q+4)$ we have $8/\eta \leq 16L(q+4)$. If $\varepsilon^2 T \leq 1/q^3$ then $\varepsilon^2 q^2 T \leq q/T$. Insert these constants into (D.4) to recover the bound stated in Theorem 1. $\qquad\square\qquad\qquad\qquad\qquad\qquad\square$

## B.3 Algorithm and hyperparameter settings

Table 6: The hyperparameters setting in our experiments.

| Experiment | Hyperparameters | Values |
|---|---|---|
| FT | Batch size | 8 |
| | Learning rate | {1e-5, 5e-5} |
| | Lr schedule | Constant for RoBERTa; Linear for OPT and LLaMA |
| MeZO | Batch size | {64, 16} |
| | Learning rate $\eta$ (Lr) | {1e-6, 5e-7} |
| | $\epsilon$ | 1e-3 |
| | Lr schedule | Constant for RoBERTa; Linear for OPT and LLaMA |
| MeZO LoRA | Batch size | {64, 16} |
| | Learning rate $\eta$ (Lr) | {1e-4, 5e-5} |
| | $\epsilon$ | 1e-2 |
| | Lr schedule | Constant for RoBERTa; Linear for OPT and LLaMA |
| P-GAP | Batch size | {64, 16} |
| | Learning rate $\eta$ (Lr) | {2e-4, 1e-4, 5e-5} |
| | $\epsilon$ | 1e-2 |
| | Window size $k$ | 100 |
| | Number of probe perturbations $h$ | 10 |
| | Rank $r$ | {128,256,512} |
| | Projection magnitude $\delta$ | Initialized as 2 and gradually decayed it to 0 |
| P-GAP (LoRA) | Batch size | {64, 16} |
| | Learning rate $\eta$ (Lr) | {3e-2, 5e-2, 1e-2} |
| | $\epsilon$ | 1e-1 |
| | Window size $k$ | 100 |
| | Number of probe perturbations $h$ | 10 |
| | Rank $r$ | {8} |
| | Projection magnitude $\delta$ | Initialized as 2 and gradually decayed it to 0 |

---

**Algorithm 1** Corrected Projected Gradient Directions with Low-Dimensional Perturbations (Lazy ZO for LLMs)

---

**Require:** Parameters $\boldsymbol{\theta}$, dataset $\mathcal{D}$, window size $k$, number of probe perturbations $h$, rank $r$, perturbation scale $\varepsilon$, learning rate $\eta$, projection magnitude $\delta$, loss function $\mathcal{L}$, iteration steps $T$, set of all matrices needed to be fine-tuned $\mathcal{M}$

1: $t \leftarrow 0$
2: **while** $t \leq T$ **do**
3:     **if** $t \bmod k = 0$ **then**
4:         $(\{\boldsymbol{U}_r^\ell, \boldsymbol{S}_r^\ell, \boldsymbol{V}_r^\ell\})_{\ell \in \mathcal{M}} \leftarrow \text{LOWERDIMGENERATE}(\boldsymbol{\theta}, h, r, \varepsilon)$
5:     **end if**
6:     **for all** parameter $\boldsymbol{W}_\ell \in \boldsymbol{\theta}$ **do**
7:         **if** $\boldsymbol{W}_\ell$ is matrix and $\ell \in \mathcal{M}$ **then**
8:             Sample $\mathcal{Z}_{init} \sim \mathcal{N}(0, I_{r \times r})$
9:             $\mathcal{Z} \leftarrow \text{PROJECTION}(\mathcal{Z}_{init}, \boldsymbol{S}_\ell^r, \delta)$          $\triangleright \langle \boldsymbol{S}_\ell^r, \mathcal{Z} \rangle_F = \xi \sqrt{\delta} \|\boldsymbol{S}_\ell^r\|_F$
10:             $\mathcal{Z}_f \leftarrow \boldsymbol{U}_r^\ell \mathcal{Z} (\boldsymbol{V}_r^\ell)^T$
11:         **else**
12:             Sample $\mathcal{Z}_f \sim \mathcal{N}(0, I)$
13:         **end if**
14:     **end for**
15:     $\mathcal{L}_+ \leftarrow \mathcal{L}(\theta + \varepsilon z), \quad \mathcal{L}_- \leftarrow \mathcal{L}(\theta - \varepsilon z)$
16:     $\mathcal{G}_t \leftarrow (\ell_+ - \ell_-)/(2\varepsilon)$
17:     **for all** $\boldsymbol{W}_\ell \in \theta$ **do**
18:         $\boldsymbol{W}_\ell \leftarrow \boldsymbol{W}_\ell - \eta \, \mathcal{G}_t \, \mathcal{Z}_f$
19:     **end for**
20:     $t \leftarrow t + 1$
21: **end while**

---

We have provide the computational process of P-GAP in the **Algorithm 1**. As discussed in our analysis of variance in Appendix B.1, the reduction in the number of perturbed parameters necessitates corresponding adjustments to both the learning rate $\eta$ and the perturbation scale $\epsilon$. The specific choices of learning rate $\eta$ and perturbation scale $\epsilon$ used in our experiments are detailed in Table 6. In our experiments, we found that the projection magnitude $\delta$ can be set relatively large at the beginning of training and then gradually reduced in the later stages. This strategy leads to better final performance and improved convergence efficiency. Therefore, in practice, we initialized the projection magnitude $\delta = 2$ and gradually decayed it to 0 as the training progressed. Moreover, we set $k = 100$ and $h = 10$ in all of our experiments.

---

**Algorithm 2** LOWERDIMGENERATE($\boldsymbol{\theta}, h, r, \varepsilon$)

---

**Require:** Current parameters $\boldsymbol{\theta}$, number of probe perturbations $h$, rank $r$, step size $\varepsilon$
1: **for all** matrix parameter $\boldsymbol{W}_\ell, \ell \in \mathcal{M}$ **do**
2:     $\boldsymbol{G}_\ell \leftarrow 0$
3: **end for**
4: **for** $j = 1$ to $h$ **do**
5:     Sample $\boldsymbol{Q}_\ell^j$ with each $\boldsymbol{Q}_\ell^j \sim \mathcal{N}(0, I)$
6:     $\mathcal{L}_+^j \leftarrow \mathcal{L}(\boldsymbol{\theta} + \varepsilon \boldsymbol{Q}_\ell^j); \; \mathcal{L}_+^j - \leftarrow \mathcal{L}(\boldsymbol{\theta} - \varepsilon \boldsymbol{Q}_\ell^j)$
7:     $\rho \leftarrow (\mathcal{L}_+^j - \mathcal{L}_-^j)/(2\varepsilon)$
8:     **for all** matrix $W_\ell$ **do**
9:         $\boldsymbol{G}_\ell \leftarrow \boldsymbol{G}_\ell + \frac{\rho}{h} \boldsymbol{Q}_\ell^j$
10:     **end for**
11: **end for**
12: **for all** matrix $\boldsymbol{W}_\ell$ **do**
13:     $(\boldsymbol{U}_r^\ell, \boldsymbol{S}_r^\ell, \boldsymbol{V}_r^\ell) \leftarrow$ svd_lowrank($\boldsymbol{G}_\ell, q = r$)
14: **end for**
15: **return** $(\boldsymbol{U}_r^\ell, \boldsymbol{S}_r^\ell, \boldsymbol{V}_r^\ell)_\mathcal{M}$

---

**Algorithm 3** PROJECTION($\mathcal{Z}_{init}, \boldsymbol{S}_\ell^r, \delta$)

---

**Require:** Initial $\mathcal{Z}_{init} \in \mathbb{R}^{r \times r}$; coefficient matrix $\boldsymbol{S}_\ell^r$; projection magnitude $\delta$
**Ensure:** We want to get the projected parallel component $\mathcal{Z}$ of $\mathcal{Z}_{init}$ such that $\langle \boldsymbol{S}_\ell^r, \mathcal{Z} \rangle_F = \xi \sqrt{\delta} \|\boldsymbol{S}_\ell^r\|_F$, with $\xi \in \{-1, 1\}$

1: $\xi \sim \text{Uniform}\{-1, 1\}$
2: $f \leftarrow \langle \boldsymbol{S}_\ell^r, \mathcal{Z}_{init} \rangle_F, \quad g \leftarrow \|\boldsymbol{S}_\ell^r\|_F$
3: $\alpha \leftarrow \dfrac{f - \xi \sqrt{\delta}\, g}{g^2 + 10^{-12}}$
4: **return** $\mathcal{Z} \leftarrow \mathcal{Z}_{init} - \alpha\, \boldsymbol{S}_\ell^r$

---

### B.4 MORE RESULTS

We also provide fine-tuning experiments of KerZOO on the LLaMA-3 model series. Hyperparameters are generally the same with OPT series models fine-tuning. The detailed results of the experiments are shown in Table 7 and 8 below.

We further evaluate the training efficiency and memory footprint of P-GAP on the OPT-2.7B model across SST-2 and RTE. Compared with MeZO and HiZOO, P-GAP achieves a better balance between memory usage and convergence speed. On both datasets, P-GAP substantially reduces training time while keeping the memory cost within a moderate increase compared to MeZO but less than HiZOO. In particular, when combined with LoRA on RTE, P-GAP+LoRA consumes less than 20% of the training time of MeZO, yet maintains competitive performance. These results highlight that P-GAP can serve as an efficient and scalable alternative for large-scale fine-tuning.

Table 7: Experiment results on LLaMA3-3B (1000 training samples)

| Task | SST-2 | RTE | CB | WSC | WIC |
|------|-------|-----|-----|-----|-----|
| FT | 94.2 | 81.2 | 91.4 | 72.2 | 63.8 |
| MeZO | 89.0 | **63.8** | 69.6 | 62.5 | 58.2 |
| P-GAP | **92.3** | **63.8** | **73.2** | **64.6** | **59.8** |

Table 8: Experiment results on LLaMA3-8B (1000 training samples)

| Task | SST-2 | RTE | CB | WSC | WIC |
|------|-------|-----|-----|-----|-----|
| MeZO | 91.2 | 61.0 | 73.2 | 64.4 | 59.2 |
| P-GAP | **93.0** | **67.2** | **75.0** | **65.8** | **60.2** |

Table 9: Memory and training time comparison of OPT-2.7B on SST-2 dataset (35 tokens per example on average)

| Method | Memory cost | Iteration step | GPU hours |
|--------|-------------|----------------|-----------|
| FT | 45.4G | 9.3% | 16.8% |
| LoRA | 18.5G | 5.6% | 4.3% |
| MeZO | 6.8G | 100.0% | 100.0% |
| HiZOO | 11.3G | 59.2% | 87.4% |
| P-GAP | 8.7G | 34.9% | 68.0% |
| MeZO+LoRA | 5.5G | 74.1% | 43.7% |
| HiZOO+LoRA | 5.7G | 46.3% | 41.0% |
| P-GAP+LoRA | 5.9G | 34.7% | 29.9% |

Table 10: Memory and training time comparison of OPT-2.7B on RTE dataset (180 tokens per example on average)

| Method | Memory cost | Iteration step | GPU hours |
|--------|-------------|----------------|-----------|
| FT | 62.2G | 10.0% | 16.2% |
| LoRA | 42.5G | 8.3% | 6.6% |
| MeZO | 7.8G | 100.0% | 100.0% |
| HiZOO | 13.2G | 63.3% | 88.9% |
| P-GAP | 10.5G | 24.5% | 64.1% |
| MeZO+LoRA | 7.5G | 73.3% | 34.8% |
| HiZOO+LoRA | 7.8G | 56.7% | 35.9% |
| P-GAP+LoRA | 7.6G | 16.9% | 8.7% |

