# OpenReview forum: "Towards Fast LLM Fine-tuning through Zeroth-Order Optimization with Projected Gradient-Aligned Perturbations"
_ICLR.cc/2026/Conference — ICLR 2026 Conference Withdrawn Submission_

### Official Review · Reviewer_yw6x · 2025-10-18

**Soundness:** 3
**Presentation:** 2
**Contribution:** 2
**Rating:** 4
**Confidence:** 4

**Summary:**

This paper proposes P-GAP, a zeroth-order optimization method for LLM fine-tuning that reduces gradient estimation variance by projecting perturbations into a low-dimensional subspace and aligning them with approximate gradient directions. The method is motivated by efficiency constraints in large-scale fine-tuning without backpropagation and aims to improve convergence speed and memory efficiency.

**Strengths:**

- The proposed projection-based, gradient-aligned perturbation design is useful for this task.

- The paper provides thorough theoretical analysis to support the method.

**Weaknesses:**

- The paper lacks sufficient ablation studies on hyperparameters (e.g., subspace dimension, update frequency, perturbation scale). How sensitive is the method to these choices, and what guidelines should practitioners follow?

- Table 3 and Table 4 only report partial experiments. Could the authors provide results on more datasets to confirm robustness and generality?

- The experiments mainly focus on OPT models, which are relatively outdated. It would strengthen the claims to include newer LLMs to validate the generalization ability of the proposed method.

- The writing feels rushed. Important algorithm are only in the appendix; maybe these should be brought into the main text for clarity and accessibility.

**Questions:**

Seen in weakness

---

### Official Review · Reviewer_pUhR · 2025-10-29

**Soundness:** 2
**Presentation:** 1
**Contribution:** 2
**Rating:** 2
**Confidence:** 3

**Summary:**

This paper proposes P-GAP (Projected Gradient-Aligned Perturbation), a new  fine-tuning method for LLM based on zeroth-order (ZO) optimization. P-GAP reduces the variance of ZO gradient estimation by finding proper perturbation directions in the space of high-dimensional matrices and further decreases the parameter space through low-rank decomposition. Specifically, it performs SVD to estimate a low-dimensional gradient subspace and aligns Gaussian perturbation in that subspace. Empircal experiments on models including RoBERTa-large, OPT 2.7B/6.7B/13B, and LLaMA 3B/8B show that P-GAP achieves faster convergence and higher accuracy compared to baselines across diverse classification and generation benchmarks.

**Strengths:**

The paper presents an interesting extension of directionally aligned perturbations from vectors to matrices to reduce variance in ZO gradient estimation. The authors conduct comprehensive experiments to demonstrate the effectiveness of the proposed method.

**Weaknesses:**

The paper argues that P-GAP reduces variance through low-dimensional perturbation spaces, but there are no explicit variance measurements or visualizations supporting this claim. The main algorithm (Algorithm 1) is only briefly referenced in the main text, and the appendix lacks a detailed explanation of their theoretical results.

**Questions:**

- The authors claim that working in a low-dimensional space helps reduce variance, but no empirical evidence is provided to support this claim. How can we be sure that the  performance improvements in the experiments are due to variance reduction rather than other factors? Since the central motivation of the paper is variance reduction, the authors should provide more detailed evidence to support this claim.

- The appendix is just a list of technical theoretical statements without any explanations. It is difficult to read and follow, and additional context or intuition would help readers better understand the results.

- Algorithm 1 introduces many hyperparameters but the paper provides little explanations about how to choose them in practice. The authors should include practical guidance or sensitivity analysis to help practitioners set these hyperparameters effectively.

- The descriptions of the algorithms have several notation errors. For example, what does $S_{\ell}^{r}$ in eq (15) represent? The notation used in Algorithm 1 and Algorithm 2 is inconsistent (e.g., $S_{\ell}^{r}$ and $S_{r}^{\ell}$, $\ell_+, \ell_-$ in line 16). What is $z$ in line 15?

- Typo in line 6 of Algorithm 2 in the expression $\mathcal{L}^j_+ -$.

- In line 12, if the parameter is not a matrix, does the algorithm just perform standard ZO gradient estimation?

---

### Official Review · Reviewer_gySz · 2025-11-01

**Soundness:** 2
**Presentation:** 2
**Contribution:** 2
**Rating:** 2
**Confidence:** 5

**Summary:**

The paper proposes **P-GAP**, a zeroth-order (ZO) fine-tuning scheme for LLMs that samples perturbations in a **low-dimensional gradient subspace** and then **aligns** them with a projected gradient direction. The subspace is obtained via a low-rank factorization (e.g., SVD) and is updated lazily to amortize cost; ZO gradient estimates are then formed using corrected low-dimensional perturbations mapped back to the full parameter space. The authors provide variance and convergence analyses arguing that restricting/aligning perturbations lowers estimator variance relative to standard ZO. They evaluate on encoder (RoBERTa-large) and decoder models (OPT/LLaMA) across classification and QA tasks, and report faster convergence, reduced GPU hours, and higher accuracy compared to ZO baselines; they also report results with LoRA and include wall-clock and memory analyses.

**Strengths:**

* Clearly targets the known high-variance issue of ZO by combining dimensionality reduction with directional alignment of perturbations.
* The projection/alignment step is simple, uses standard linear algebra, and can be dropped into existing ZO pipelines (incl. PEFT/LoRA) with minimal changes.
* Provides analysis on variance scaling with dimensionality and argues why the proposed projection reduces estimator variance; includes a convergence discussion under stated assumptions.
* Uses a lazy-update strategy to limit subspace estimation overhead; focuses on wall-clock and memory implications that matter for resource-constrained fine-tuning.
* Evaluations span multiple model families and parameter scales; the paper reports gains over representative ZO baselines and  reports additional improvements when combined with LoRA.

**Weaknesses:**

1. **Near-duplicate figures/tables vs. #12282.** The plotting/table template and ordering are *almost identical* to **#12282**, with colors changed: **#12350 Fig.3 / Fig.2 / Fig.4 ≈ #12282 Fig.1 / Fig.2 / Fig.3** (axes, legends, and layouts closely match).

2. **If shared templates are acceptable, how to explain drifting baselines?** Under ostensibly comparable settings, baselines diverge between the two papers in ways that **systematically favor each paper’s method**.

   * Example (**Table 2 · DROP**): **#12350** MeZO-LoRA=19.2 / HiZOO-LoRA=18.3 / **P-GAP-LoRA=22.5** **vs.** **#12282** MeZO-LoRA=13.4 / HiZOO-LoRA=13.9 / KerZOO-LoRA=14.7.
   * Example (**Table 3**): similar column-wise discrepancies under matched configurations.

3. **Even within this paper, baselines are modified without explanation.** Several MeZO entries (and MeZO-LoRA in particular) differ from the MeZO paper, while others (e.g., the **Zero-shot** row) track the original exactly. The paper does not state which baselines were re-run vs. quoted, nor justify the changes—differences consistently benefit the proposed method.

4. **Reproducibility gap.** The code release lacks **requirements.txt / environment specs, hyperparameters and seeds, preventing independent reproduction and verification of the baseline inconsistencies.

**Questions:**

See weakness

---

### Note · Authors · 2025-11-26

I have read and agree with the venue's withdrawal policy on behalf of myself and my co-authors.